# Detection of traces of calcium oxalate druses in fossil leaves of angiosperms and gymnosperms from different sites and geological periods

Mahdieh Malekhosseini[1]*, Hans-Jürgen Ensikat[1], Victoria E. McCoy[2], Torsten Wappler[1,3], Jes Rust[1]

1 Bonn Institute of Organismic Biology, University of Bonn, Bonn, Germany, 2 Department of Geosciences, University of Wisconsin-Milwaukee, Milwaukee, Wisconsin, United States of America, 3 Natural History Department, Hessisches Landesmuseum Darmstadt, Darmstadt, Germany

* mmalekho@uni-bonn.de

## Abstract

Calcium biomineralization in plants occurs in a variety of patterns such as calcium carbonate cystoliths and calcium oxalate (CaOx) crystals and agglomerates in different forms. CaOx druses and prismatic crystals with sizes between 20–100 μm are found in large amounts in the leaves of many extant plants, in angiosperms particularly in dicotyledons. In gymnosperms, large CaOx druses are often found in Cycadales and Ginkgo along the leaf veins, while most conifers contain microcrystals of <10 μm size in the parenchyma. In plant fossils, patterns of calcium biominerals are rarely reported because they usually disappear during fossilization. Traces of CaOx druses have been reported recently in fossils of dicotyledon plant leaves from Oligocene; here the CaOx was replaced by organic or mineral compounds. But there is still no certain report of CaOx druses traces in Paleozoic or Mesozoic fossils. In the study presented here, granular structures in fossil leaves from different sites across the Devonian to the Neogene were investigated and compared with biomineral patterns in extant leaves of gymnosperm and angiosperm trees. These granular structures resembled patterns of CaOx druses in extant leaves in morphology and distribution and were interpreted as probable casts of CaOx druses. Well-preserved angiosperm fossils from various sites such as seed ferns since Devonian, and Ginkgophytae since Carboniferous all showed such granular traces. The diverse chemical composition of these casts of CaOx druses (e.g., pyrite, iron oxide, organic material, $SiO_2$) depends on fossilization conditions and the chemistry of the surrounding matrix. Good knowledge of the morphology and distribution patterns of biominerals in all relevant plant groups is a basic prerequisite for recognizing their traces in plant fossils. This first extensive study of previously overlooked traces of CaOx druses in plant fossils is a promising step toward a more detailed identification of these fossil microstructures.

**Data availability statement:** All relevant data are located at Zenodo: https://doi.org/10.5281/zenodo.17477143.

**Funding:** The author(s) received no specific funding for this work.

**Competing interests:** The authors have declared that no competing interests exist.

## Introduction

Calcium oxalate (CaOx) crystals and crystal agglomerates in form of druses and raphide bundles are common in the majority of plants. They are estimated to occur in approximately 75% of angiosperm species [1,2] and can be found in almost all parts of plants. For gymnosperms, we have not found comprehensive statistics, but we detected it in all (20 out of 20) conifer and cycad species we have examined so far. In extant leaves of many species, particularly dicotyledones, CaOx crystals present different morphologies, which can include raphides, druses, styloids, prisms, and crystal sand. The most common ones are druses, prismatic crystals, and raphides [3]. CaOx druses, also known as spherulites, are globular aggregates of crystals growing from a common center and occur individually in a cell. In leaves of many species, particularly dicotyledones, CaOx crystals and druses occur in remarkable diverse distribution patterns (macro patterns), partly associated with the venation, randomly distributed in the mesophyll, or at other specific locations (e.g., in or under the epidermis), which may be linked to specific functions such as defense or effect on the intensity of solar radiation [4]. A high diversity can be found even within single plant families, such as Cactaceae [5,6], where almost all morphological types have been found. The size of fully-grown druses and solitary crystals varies largely from < 5 μm to more than 100 μm length or diameter, typically from 20 to 80 μm for druses. The crystals are formed mainly in the intra-vacuolar chamber of specialized cells, which are known as idioblasts [7]. The actual mechanisms that regulate the shape of the CaOx crystals have not yet been elucidated [3].

Leaves of most extant common dicotyledoneous trees, which were subjects of Malekhosseini et al. (2022) [8], contained CaOx druses and/or prismatic crystals in various combinations of size (mainly 20–80 μm), concentration (up to 12% of the leaf dry mass; own data, not yet published), and location either in the mesophyll or attached to the venation (Fig 1). Raphides were very rarely found in leaves of dicoty-ledoneous trees or shrubs, sometimes found in small herbs, and commonly found in certain groups of monocotyledons such as Araceae and palms [9]. Other monocots such as grasses (Poaceae) usually contain no CaOx but are often mineralized with silica [9].

Extant gymnosperms show two fundamental types of CaOx depositions. Cycadales and *Ginkgo* leaves contain intracellular druses with sizes up to 100 μm, sometimes mixed with crystal sand (Fig 2A–2H). In contrast, conifers (Pinopsida) contain extracellular depositions of very small CaOx crystals (usually < 5 μm) on the surfaces of parenchyma cells and embedded in epidermal outer cell walls.

In contrast to their often striking depositions in fresh leaves, fossil leaves usually do not contain any remnants of calcium biominerals, due to the obvious instability of CaOx and $CaCO_3$ during fossilization of plants. (Calcium oxalate is susceptible to oxidation to carbonate; calcium carbonate is soluble even in very weak acids.) Reports on traces of CaOx druses in fossil leaves were lacking until recently when we identified regular patterns of granular structures as traces and casts of CaOx druses in well-preserved fossil leaves from the Oligocene Lagerstätte Rott (Bonn, Germany) [8]. Supposedly, these structures had been observed frequently, but were

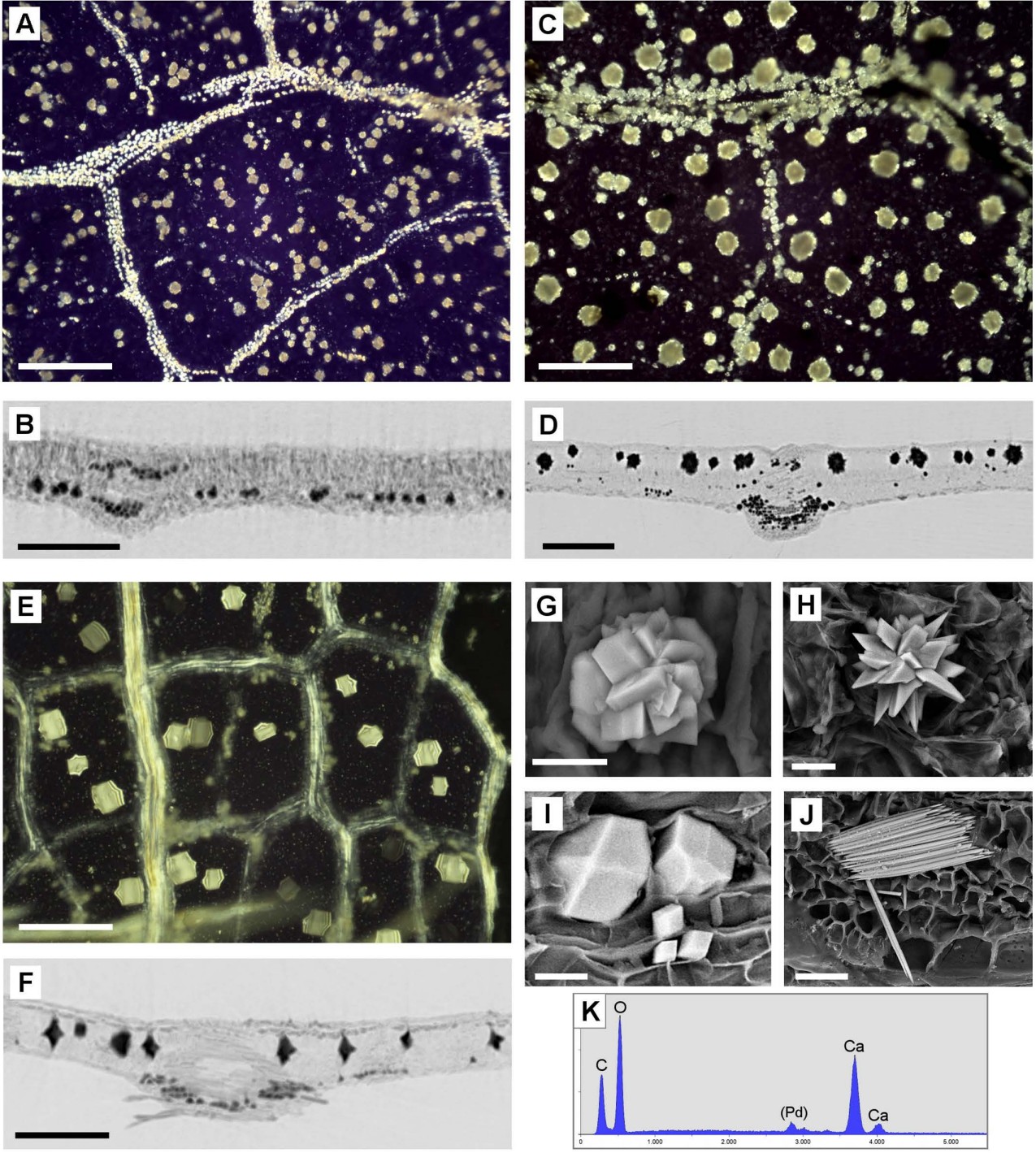

**Fig 1. Distribution and morphology of CaOx crystals in leaves of selected angiosperms (dicotyledons).** (A, C, E): Pol-LM images of inciner-ated (A, C) or chemically cleared (E) leaves show the distribution and sizes of druses and crystals in the leaf lamina and along the leaf veins. (B, D, F): Micro-CT images; reconstructed cross sections show the three-dimensional distribution within the leaves. (A, B): *Quercus robur* with small (20 μm) druses in aerenchyma and small crystals along veins. (C, D): *Juglans regia* with large (up to 80 μm) druses between pallisade parenchyma and small druses in aerenchyma and along veins. (E, F): *Carpinus kawakamii* with very large (up to 100 μm) crystals, reaching from upper to lower epidermis, and small crystals along veins. (G-J): SEM images of common CaOx crystal types. (G): Compact druses, in *Prunus avium*; (H): Fragile druses with sharp tips, in *Conocarpus erectus*; (I): Prismatic crystals, in *Parrotia persica*; (J): needle-shaped raphides, in *Nannorrhops ritchiena* (Arecaceae, palm). (K): EDX spectrum of CaOx crystal in (I); Scale bars: (A-F) = 200 μm; (G, H, I) = 10 μm; (J) = 20 μm.

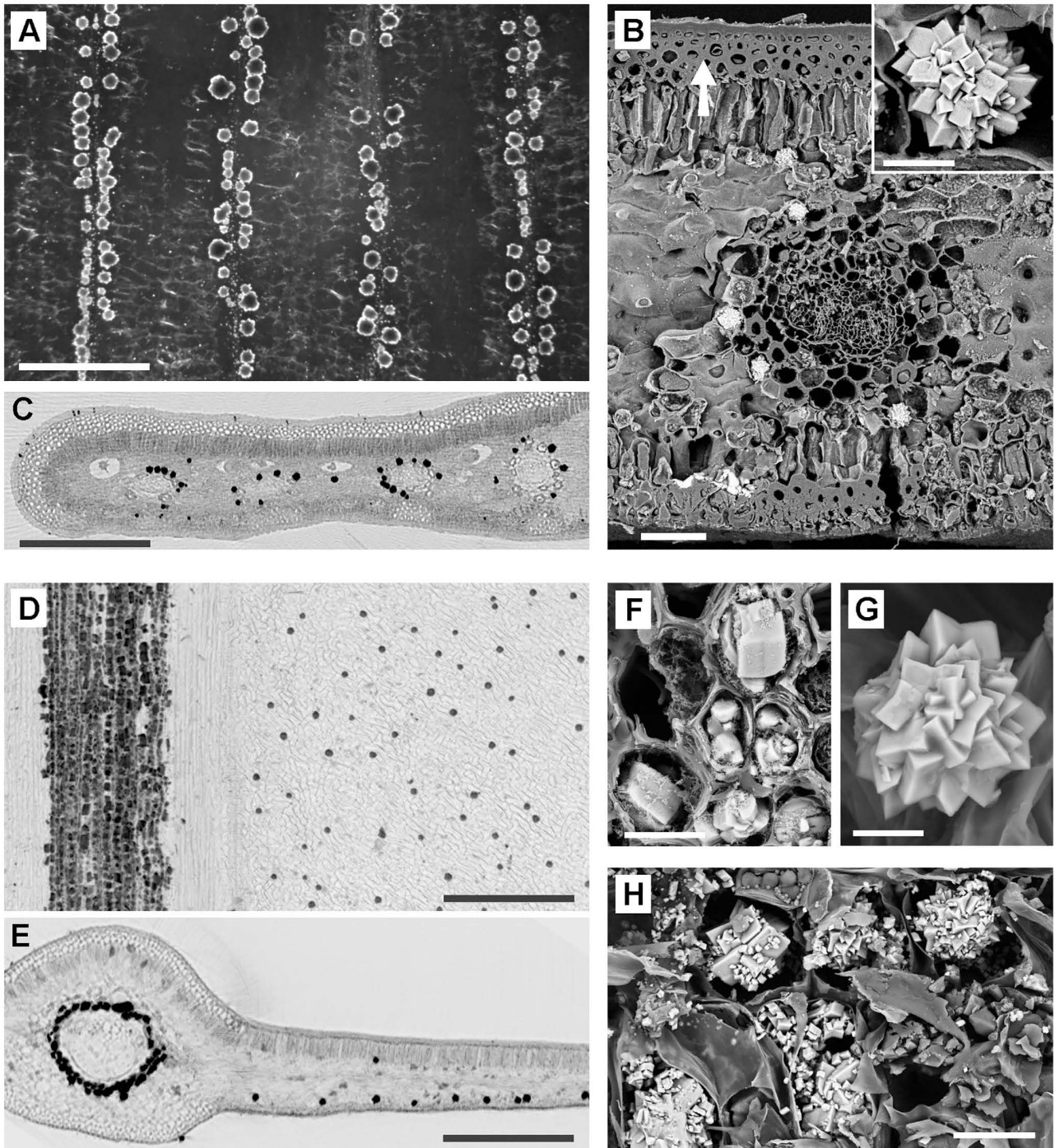

**Fig 2. Distribution and morphology of CaOx in leaves of various extant gymnosperms.** (A-C): *Encephalartos lehmannii*. Parallel veins are associated with druses. (A): Pol-LM image of incinerated leaf. (B): SEM image of druses in cross section of the leaf. Inset shows a druse in detail. Note the thick cell walls of multiple cell layers in epidermis/hypodermis (arrow). (C): μ-CT image; reconstructed cross section shows distribution of druses around veins in 3D. (D-G): Cycas *szechuanensis*. The central veins of each leaflet are densely surrounded by druses; the leaf lamina contains druses in random distribution. (D, E): μ-CT images; reconstructed plane view and cross section. (F-G): Detailed SEM images of CaOx show mix of crystals, druses, and crystal sand along veins (F), and regular druses leaf lamina (G). (H): *Cycas cyratozamia*; SEM image of parenchyma cells filled with druses and crystal sand. All SEM images are ‚compositional contrast' (BSE) images. Scale bars: (A, C, D, E) = 500 μm; (B) = 100 μm; Inset in (B) = 20 μm; (F, H) = 30 μm; (G) = 20 μm.

misinterpreted, e.g., as pollen, or not considered at all [10,11]. Furthermore, traces of small calcium oxalate crystals have been mentioned in fossil leaves of gymnosperms [12]. These casts have been interpreted as impressions of CaOx crystals in the leaf epidermis. Traces of CaOx druses can be identified reliably due to their almost globular shape and their characteristic distribution patterns (e.g., within veins or regularly distributed in mesophyll) [8]. It has to be accentuated, that morphology, size, density and distribution pattern are verifiable characters of fossil CaOx druses, while their infill is a specific feature of the individual site and a consequence of the local fossilization process.

Our previous studies have shown that well-preserved casts of CaOx druses were found only in a small fraction of leaf fossils. Main reasons are poor preservation and absence of large druses in original leaves of many species. Only a fraction of plant species contain large druses in sufficient concentration and characteristic distribution patterns, which are easily recognized when fossilized (e.g., Juglandaceae, *Quercus*, *Ginkgo*). Many other species contain only small amounts of CaOx or many small crystals. Casts of CaOx druses in fossils may be difficult to recognize and are easily confused with structures of their environment, e.g., grainy structures of crystalline sediments. In many leaf fossils, remnants of parenchyma are lost, as they can be very fragile or fragmented. The casts or cavities of CaOx in fossil leaves showed sizes and distribution patterns that resembled those of druses in fresh leaves, but they were highly variable in composition of the infill, e.g., patches of iron oxides, spherical organic granules or framboidal pyrite [8].

Although investigations of the fossilization, alteration, and replacement of CaOx biominerals in fossil plants are still needed, Malekhosseieni et al. [8] have developed a hypothesis for a general scenario of this process: the original CaOx crystals or druses have been lost during the fossilization process, leaving behind empty cavities which are sometimes refilled by organic material or inorganic minerals. Many factors such as the physical and chemical environment during fossilization, structure and composition of the sediment containing the fossils, physiology of the leaves (e.g., thickness of the leaves and position of CaOx minerals in the leaves), and the grade of metamorphosis play a major role in the fossilization of calcium minerals (e.g., [3]).

Until now, traces of CaOx druses have only been identified from the Oligocene fossil site Rott [8]; so it appeared necessary to extend the search across fossil collections from other locations and time periods, in order to achieve more insights in the fossilization processes and preservation of biomineral traces. The lack of fossil records limits our understanding of the evolution of CaOx and associated ecological behaviour in plants. The widespread occurrence and high abundance of CaOx druses in extant cycads, which are regarded as the closest relatives of extinct seed ferns and in *Ginkgo biloba* promise a high likelihood of finding their traces in gymnosperm fossils [13,14]. In the present study we examined a variety of fossil leaves from a range of fossil sites from the Devonian to the Eocene to advance a better understanding of the distribution pattern of CaOx biominerals in fossil plants. We gave special consideration to granular structures, presumed casts of former CaOx druses, in gymnosperm fossils, particularly Ginkgophyta and seed ferns, to show whether CaOx mineralization can be detected from deep time to today.

The investigation presented here endeavors to answer the following questions: (1) Do granular structures or casts of Ca-biominerals occur in older fossils, or are they limited to unique fossil sites or younger periods of time? (2) What are the main organic or inorganic elements in fossil leaves and casts of CaOx after fossilization? (3) Does the replacement of original CaOx crystals and druses follow specific patterns in different time periods or within clades?

## Materials and methods

### Material

Fossil leaves from different localities and geological periods were investigated. The material derived from several institutes and museums (Table 1). All necessary permits were obtained for the described study, which complied with all relevant regulations. The fossil collection of the Goldfuß-Museum, University of Bonn, Germany, was available for more detailed studies and preparations. The fossil samples from the available collections (Table 1) showed varying preservation quality of the leaves. Only samples with at least a semblance of granular structures have been selected for loan from

**Table 1. Fossil leaves examined from different times and sites.**

| Fossil site (Collection) | Age | Fossil species | Key Publications |
|---|---|---|---|
| Bear Island (Goldfuß Museum, University of Bonn) | Late Devonian 370 Ma | *Archaeopteris roemeriana* | Schweitzer, 2006 [15] |
| | | *Cyclostigma kiltorkense* | |
| Upper Silesian, Poland, (Col. Andrzej Gorski, 2022) (Goldfuß Museum, University of Bonn) | Carboniferous, Westfalium „B", 310 Ma | *Paripteris gigantea* | Šimůnek, 2010 [16] |
| Upper Silesia, Poland, (Col. Maciej Kania, 2024) (Goldfuß Museum, University of Bonn) | Carboniferous, Westfalium B, 310 Ma | *Eusphenopteris striata* | Josten & Amerom, 1999 [17] Gothan, 1913 [18] |
| | | *Linopteris sp.* | Josten & Amerom, 1999 [17] Zeiller, 1899 [19] |
| | | *Lonchopteris rugosa* | Brongniart, 1828 [20] |
| | | *Neuropteris rarinervis* | Josten & Amerom, 1999 [17] |
| Dunedoo, Australia, (Goldfuß- Museum, University of Bonn) | Upper Permian Ca. 275 Ma | *Glossopteris sp.* | Holmes, 1977 [21] |
| Unternschreez Bayreuth, Germany (Goldfuß Museum, University of Bonn) | Triassic, Rhaetian 204 Ma | *Otozamites brevifolius* | Weber, 1968 [22] |
| Kamloops, Canada (Goldfuß Museum, University of Bonn) | Tertiary, Early Eocene 50 Ma | *Ginkgo adiantoides* | Greenwood et al., 2016 [23] |
| Eckfeld, Germany (Naturhistorisches Museum, Mainz) | Eocene 44.3 Ma | not identified | Frankenhäuser et al., 2009 [24] |
| Willershausen, Germany (Geowissenschaftliches Museum, Göttingen) | Pliocene 3 Ma | *Betula sp.* *Liriodendron tulipifera* | Ferguson & Knobloch, 1998 [25] |

the Institutions or Museums. Many fossils consisted of coarse-grained sediments, which made a recognition of traces of druses unreliable or impossible. Fortunately, a lot of fossil leaves, even Devonian and Carboniferous, consist of carbonized leaf material on a fine-grained sediment which show finest structures with microscopic details. The best chances to find traces of former CaOx druses are given in fossil leaves which contain remnants of their parenchyma structures, either carbonized or filled with mineral precipitations, because druses usually occur in the leaf parenchyma, but rarely in the epidermis. Fossilized remnants of only epidermis or cuticle rarely contain traces of druses. Most useful are freshly exposed fossils, particularly when both sides are available for examination. We studied leaves from >200 species in order to get an idea of the distribution patterns of

druses and crystals, but not for statistics. When the microscope images were consistent with the overall results, then the images of a single leaf could be sufficient. Many species were studied multiple times with different methods. However, the images show examples of druse distributions, but not complete statistics on concentration and size, due to frequent inhomogeneity of the samples.

The original data related to this study are available from the secretary of the Paleontology Section, Institute of Organismic Biology, University of Bonn.

Furthermore, for more details data are available under the following link: Zenodo: https://doi.org/10.5281/zenodo.17477143.

All plant samples collected in this study were taken from species cultivated in the Botanical Garden, Bonn. This sample collection complies with relevant institutional, national, and international guidelines and legislation. The following species appear in this study: *Carpinus kawakamii* (accession 34895); *Conocarpus erectus* (accession 48022); *Cycas cyratozamia* (accession 34591); *Cycas szechuanensis* (accession 33081); *Encephalartos lehmannii* (accession 7687); *Encephalartos*

*villosus* (accession 15870); *Ginkgo biloba* (accession 1894); *Juglans regia* (accession 9662); *Nannorrhops ritchiena* (accession 33503); *Parrotia persica* (accession 12241); *Prunus avium* (accession 25464); *Quercus robur* (accession 1887).

## Methods

The identification of CaOx casts or traces in fossil leaves relies on identifying similarities in the morphology and distribution of these traces to the morphology and distribution of the CaOx in extant leaves, and particularly extant leaves closely related to the fossils in question. The fossil and fresh samples were examined and documented with different microscopic techniques such as light microscopy (LM), scanning electron microscopy (SEM), and micro-computertomography (μ-CT).

### Microscopy-equipment

A stereomicroscope was primarily used to recognize the granular structures on the surface of the fossil leaves. For higher magnifications, a standard light microscopy (LM) with polarizing filter attachment was used (Müller optronic, Erfurt, Germany). Both microscopes were used with a Swift SC1803 microscope camera (Swift Optical Instruments, Schertz, Texas, US) with 18-megapixel resolution. A Lumix DMC-G70 photo-camera (Panasonic Corporation, Osaka, Japan) with Lumix macro-objective was used for close-up images.

Scanning electron microscopy was performed with a Tescan VEGA 4 (www.tescan.com) and a LEO 1450 SEM (Cambridge Instruments, Cambridge, United Kingdom). Both SEMs are equipped with secondary electron (SE) and backscattered electron (BSE) detectors and energy-dispersive X-ray spectrometers (EDX) for element analysis. The Tescan SEM has low-vacuum capability. SEM imaging parameters were usually acceleration voltage of 10–20 kV, working distance 10–20 mm, and high vacuum condition. Low vacuum operation was used for examination of fossil samples without metal coating and for large, outgassing samples.

μ-CT-scans and X-ray images of dry extant leaves were obtained with a SkyScan 1272 Micro-CT system (Bruker microCT, Kontich, Belgium). The images were recorded with a detector of 4,032 × 3,280 pixels with a pixel size of 1 μm, source voltage of 30 kV, source current of 200 μA, rotation steps of 0.4 deg. Visualisation of the μ-CT data was performed with ImageJ-Fiji software (https://imagej.net/software/fiji/). Reconstructed sections and plane views were rendered with the software function ‚Z Project' using ‚Max Intensity'.

### Preparation of fossil samples for LM

Most samples required careful cleaning with an air-blower, rinsing with water, or wiping off contaminations with a soft brush. If granular structures were below the surface, then the surface layer was lifted off using hot-melt glue.

### Preparation of fossil samples for SEM

Some fossil samples were small enough for direct SEM examination. After removal of dust or contaminations, a small area of interest was sputter-coated with a thin layer (ca. 15 nm) of palladium (Sputter coater Balzers SCD 040; Balzers Union, Liechtenstein); the surrounding area was capped to avoid coating. After mounting on a specimen stub the metal-coated area was electrically connected to ground. Uncoated samples could be examined in low-vacuum mode with the BSE detector and EDX only. Some of the larger fossil samples contained small loose fragments which could be extracted for SEM. The best specimens were obtained from freshly cleaved samples. SEM specimens of fossils and fresh leaves were routinely sputter-coated with 10–15 nm palladium which does not disturb EDX element analyses and compositional-contrast imaging with the BSE detector.

### Preparation of extant plant leaves

For detailed SEM images, a standard fixation of fresh leaves was used: fixation in 70% ethanol + 4% formaldehyde in water for at least 20 hrs and dehydration with ethanol, followed by critical-point (CP) drying. Internal structures were

exposed by freeze-fracturing. Therefore, samples in the ethanol-stage were frozen in liquid nitrogen and broken, followed by CP drying. For SEM, the samples were sputter-coated with 10–15 nm palladium. The same fixation and CP-drying was used for µ-CT samples. Polarization microscopy (Pol-LM) was used to illustrate the distribution pattern of CaOx druses and crystals in the leaves in a plan view. The clearest images were often obtained from the ash of burnt leaf pieces immersed in immersion oil. Therefore, fresh or dry leaf pieces were heated in a temperature-controlled furnace to ca. 650°C for few minutes until the remains turned white. The incinerated leaf remnant was carefully dropped onto a thin layer of immersion oil and examined by Pol-LM (without coverslide in order to avoid fragmentation). Alternative preparations were used as controls, such as tissue (leaf) clearing with sodium hypochlorite-based bleach, or sectioning [26].

## Results

In all fossils examined, the original CaOx was replaced by other components, making a direct chemical or mineralogical identification impossible [8]. Depending on the fossil quality, however, there was a high degree of uncertainty in the morphological interpretation, and sufficient preservational quality is a requirement of successful identification. In well-preserved fossils, morphology + distribution are the key characteristics, but with poor preserved fossils, only the distribution could be used. Fossil leaves from different geological periods were examined and analyzed for traces of calcium-based biomineralization, particularly CaOx druses or crystals. The characteristic tracks we used were particles with spherical morphology of suitable size, occurrence in the remains of leaf parenchyma, and, mainly in seed fern fossils, distribution parallel to the venation. Results from a previous study [8] and the distribution patterns of CaOx crystals in fresh leaves have been used as references for comparison. Fig 1 shows a few examples of leaves with a rich content of druses or crystals, their distribution patterns, and the detailed morphology of some CaOx druses and crystals in angiosperms. Fig 2 illustrates typical CaOx patterns in leaves of extant gymnosperms: *Ginkgo biloba* and certain cycads with parallel venation (e.g., *Encephalartos,* Fig 2A–2C) contain large druses often in close association with the parallel venation. Other cycads with a midvein in their leaflets accumulate a mix of CaOx druses, crystals, and crystal sand in cells around the vein, whereas the leaf lamina contains druses in varying concentration (Fig 2D–2H).

In the following section the fossil leaves will be explained individually for angiosperm and gymnosperm groups.

### Angiosperm collections

**Eckfeld (Coll. Mainz).** The species of the available Middle-Eocene fossil leaves from the Eckfeld Lagerstätte were not identified. The fossil samples were contaminated with a massive layer of iron oxide/hydroxide caused by storage in glycerol (Fig 3A, top left. The inset shows an SEM image.) After cleaning of the surface carefully with water and a soft brush, the leaf structure became visible (Fig 3A, bottom right). SEM examination of the sample ‚PB 2021–4-LS‘ showed that in some places the surface of the fossil leaf had been damaged by the cleaning, so that pyritic framboids (conglomerates of small $FeS_2$ crystals) under the cuticle were exposed (Fig 3B). The inserted SEM image shows a framboid in detail. EDX analyses (Fig 3C) reveal the composition of the framboids (Fe, S; red spectrum) and surrounding matrix (Si, Al, O; blue spectrum) of Spot 1 and 2 in Fig 3D. Fractured samples of fossil leaf sample ‚PB 2021–5-LS‘ showed that framboids were distributed entirely through the leaf in between two carbonized layers (arrows in Fig 3D). The occurrence of the pyrite framboids within the parenchyma remnants and their size are in accordance with the assumption that CaOx has been substituted during fossilization with available elements of the surrounding environment.

**Willershausen (Coll. Göttingen).** LM examination of several Pliocene fossil leaves from the Willershausen Lagerstätte showed a regular distribution pattern of granular structures which resembled those on angiosperm leaves from the Oligocene Rott Lagerstätte, which have been identified as casts of CaOx druses filled with either organic material or minerals [8].

SEM images of a fossil leaf assigned as *Betula sp*. (Sample ‚Göttingen 3480‘) (Fig 4A) show some of the regularly distributed globular structures with diameters of 40–60 µm. EDX analyses showed that the casts were filled with a mix of

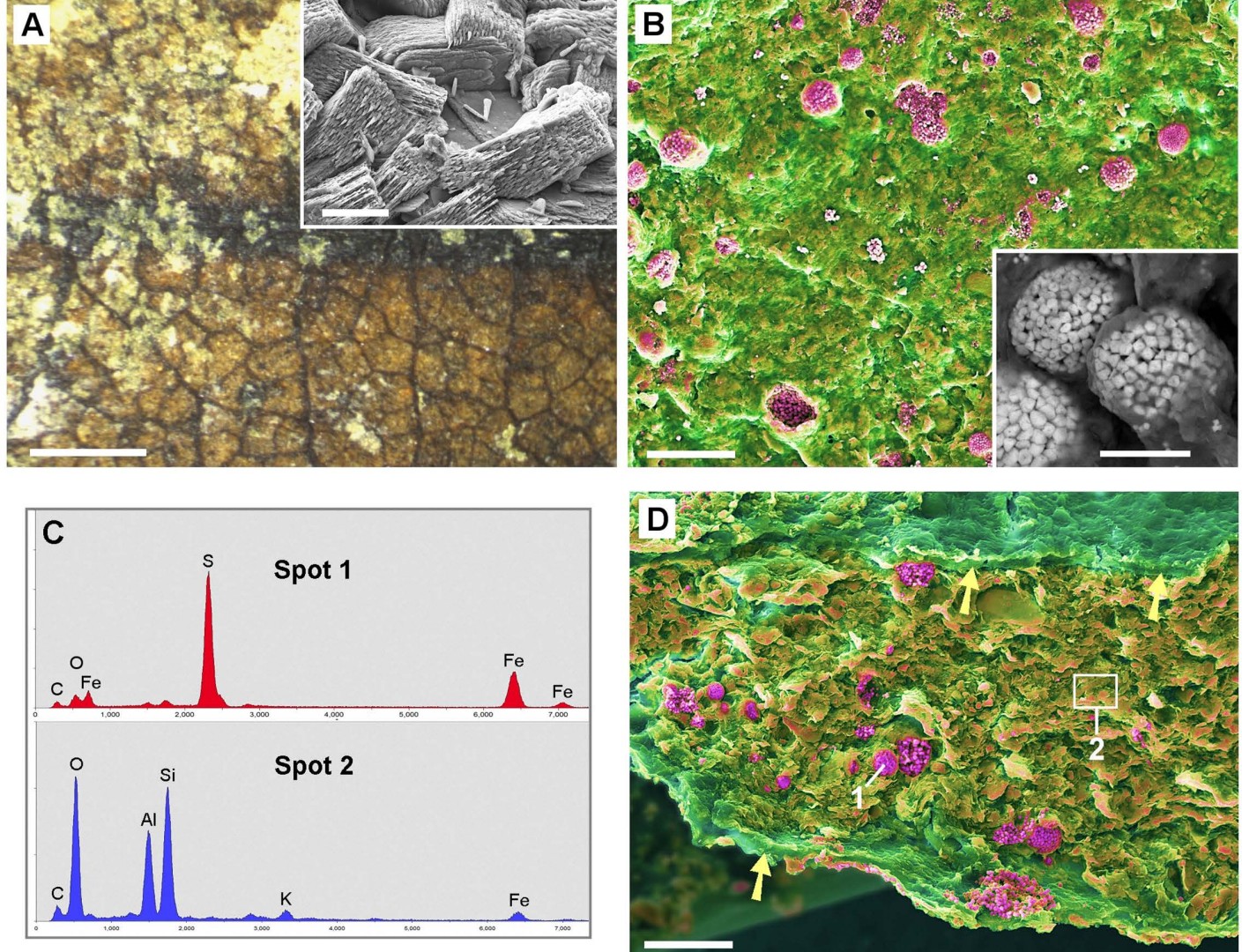

**Fig 3. Tertiary fossil leaves from the Eckfeld Lagerstätte, Middle-Eocene, with granular pyrite framboids.** (A): Stereo-microscope view of sample ‚PB 2021-4-LS' shows a partly contamination layer of iron hydroxide (top left and inserted SEM image). The leaf surface is visible after removal of the contamination (bottom right). (B-D): SEM images and analyses of Fossils ‚PB 2021-4-LS'(B) and PB 2021-5-LS' (D) from Eckfeld. Combined topographic and compositional contrast false-color images show heavy elements (Fe) in red color, Si and Al oxides in yellow-orange, organic matter (carbon) in green. (B): Surface of a slightly brushed leaf sample; pyrite framboids appear exposed under damaged cuticle layer. Inset in (A) shows framboids in detail. (C): EDX spectra show composition of framboids (Spot 1) (Fe, S; red spectrum) and surrounding material (Spot 2) (Si, Al, O; traces of Fe, K; blue spectrum) between two carbonized layers (arrows) in (D). (D): Edge of fractured sample with numerous framboids (red). Arrows indicate carbonized upper and lower epidermis walls. Scale bars: (A) = 1 mm; (B,D) = 30 µm; inset in (A) = 30 µm; inset in (B) = 10 µm.

mineral components containing Ca, Si, Al, Mg with varying composition, including granules with high contents of calcium sulfate ($CaSO_4$) e.g., in Spot 2. The matrix (Spot 3) is organic material; the EDX analysis revealed 75% C and 7% S by weight. SEM images of Sample ‚Göttingen 8518a' (assigned as *Liriodendron tulipifera*; Fig 4C–4E) show the spherical shape and variable size of organic granules. The dark appearance in the BSE image (Fig 4D) and EDX spectra indicate their organic composition; rare traces of minerals appear bright.

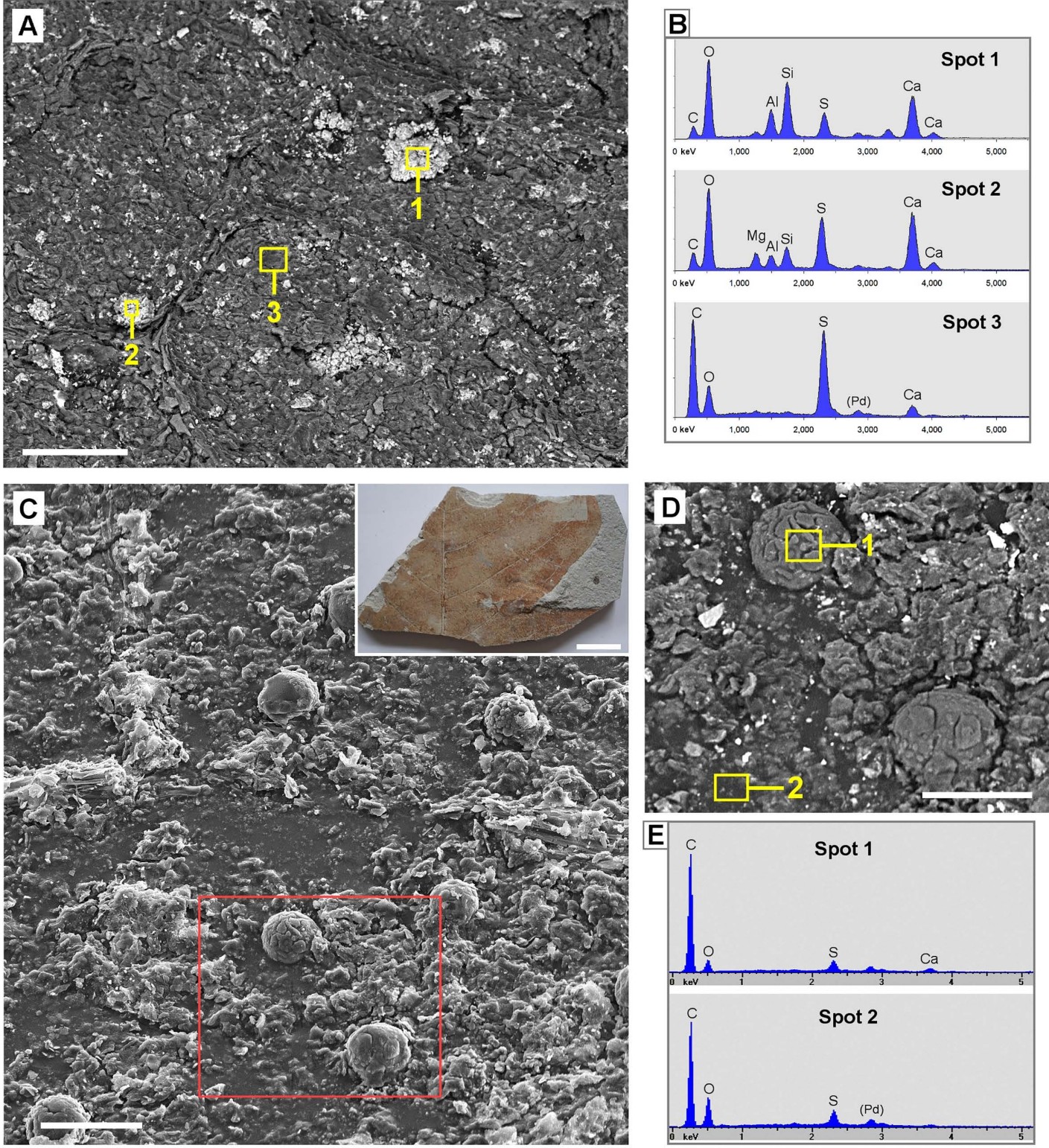

**Fig 4. Tertiary fossil leaves from Willershausen Lagerstätte with granular structures.** SEM images and EDX element analyses reveal diverse compositions of granules. (A-B): Sample ‚Göttingen 3480' (assigned as *Betula sp*.). Compositional-contrast image shows bright granules embedded in the carbonized leaf lamina. (B): EDX spectra of bright granules shows a mix of the mineral elements Ca, Si, Al, Mg, O, S. The matrix (Spot 3) consists of

organic material (mainly C and S). (C-E): Sample ‚Göttingen 8518a' (assigned as *Liriodendron tulipifera*). (C): Topography image shows globular shape of granules. Inset shows a photo of the fossil sample. (D): Compositional contrast image of marked area in (C); the dark appearance indicates organic material. (E): EDX spectra of a granular structure (Spot 1) and matrix (Spot 2) in (D) show high concentration of carbon. Scale bars: (A) = 100 μm; (C) = 50 μm; inset in (C) = 5 mm; (D) = 30 μm.

## Gymnosperm collections

**Devonian.** From a collection of several hundred samples of Late Devonian leaf fossils from Bear Island (Norway) ca. 30 specimens have been selected for a closer examination. The majority of the collection samples were assigned to *Archaeopteris roemeriana, Cyclostigma kiltorkense,* or *Pseudobornia ursina*. The Late Devonian on Bear Island is stratigraphically summarised as Røedvika Formation [27]. Fossil samples assigned to *Cyclostigma kiltorkense,* a lycopsid, consisted of carbonized black leaf remnants on a mineral substrate. Separated fragments of the leaves were covered with small yellow particles rich in Fe and S (likely pyrite) on both sides (Fig 5A, 5C), which may be casts of former leaf surface structures. A view onto the edge of the samples showed mineral inclusions in the middle of the 70–80 μm thick carbon layer (Fig 5B, 5C). In some locations, parts of the brittle carbon layer had been broken off, and granular pyrite inclusions were visible in detail, either as individual crystals or as framboids with spherical shape (Fig 5D–5E). Their size (20–30 μm) and location within the leaf, between the carbonized epidermis layers, are indications that they may be casts of former CaOx druses. Micro-CT scans were used to determine the spatial distribution of the mineral inclusions. A plan transmission view (Fig 5G) shows the entirety of granules; the reconstructed cross section (Fig 5H) shows pyrite granules (black) and Si/Al-minerals (light-gray) in the middle of the fragment. The carbon layer is transparent and almost invisible. The EDX spectrum (Fig 5F) shows the composition (Fe, S) of granules in (E).

The carbonized leaf remnants of *Archaeopteris roemeriana*, a pro-gymnosperm which are a group of early plants likely ancestral to gymnosperms among the Devonian fossils included in this study, showed a particularly rough surface structure and, in areas where the superficial leaf layer was removed, a pattern of black granules was visible (Fig 6 B, C). The size (50–80 μm) and distribution of these patches resembled those of CaOx druses in fresh leaves of gymnosperms such as *Ginkgo* or *Encephalartos* (Figs 2 A, 8 E). EDX spectra (Fig 6 D) show the compositions of the sediment (Si, Al, O; Si:Al ratio = 1.9:1) and the black grains which were organic remnants (C, O) with traces of Fe.

**Carboniferous.** Several Carboniferous leaf samples consisted of relatively thick (25–30 μm) carbon layers which easily could be separated from the sediment. SEM images of the fossil leaf of *Paripteris gigantea* (Pteridospermatophyta or ‚seed ferns') [16] show a carbonized layer (Fig 7 A). Beneath the carbon layer, a regular pattern of bright patches is visible, which are embedded in the mineral sediment and seem to be arranged along the leaf venation. EDX spectra (Fig 7 C) show the composition of granules and sediment. The patches have diameters of 30–50 μm and consist of iron oxide; the sediment consists of Al, Si, and O (Si:Al ratio ca. 1:1). Under the LM, the iron oxide inclusions appear as yellow granules (Fig 7 B, D) with a distribution pattern in parallel rows (arrows in Fig 7 D) that resembles the distribution of CaOx druses, e.g., in fresh leaves of the cycad *Encephalartos villosus* (Fig 7 E). The origin of black carbonized grains embedded in the sediment (Fig 7 A) remains speculative. They may be remnants of other leaf structures, but it seems possible that CaOx druses in close vicinity to the cuticle has been replaced by organic material from the cuticle. Other very well preserved seed fern fossils from the Upper Silesian Coal Basin show similar granular patterns in the rough leave structures, such as *Eusphenopteris striata*, *Linopteris sp.*, *Lonchopteris rugosa*, and *Neuropteris rarinervis* (Fig 8).

**Permian.** Fossil leaves of the seed fern *Glossopteris* sp. (Dunedoo, Australia; Table 1) frequently show granular structures in regions where remains of parenchymatic tissue are preserved (Fig 9 A, B; arrows). In other regions, flat round patches of ca. 30 μm diameter can be recognised by their composition or texture (Fig 9 C, D). The patches were

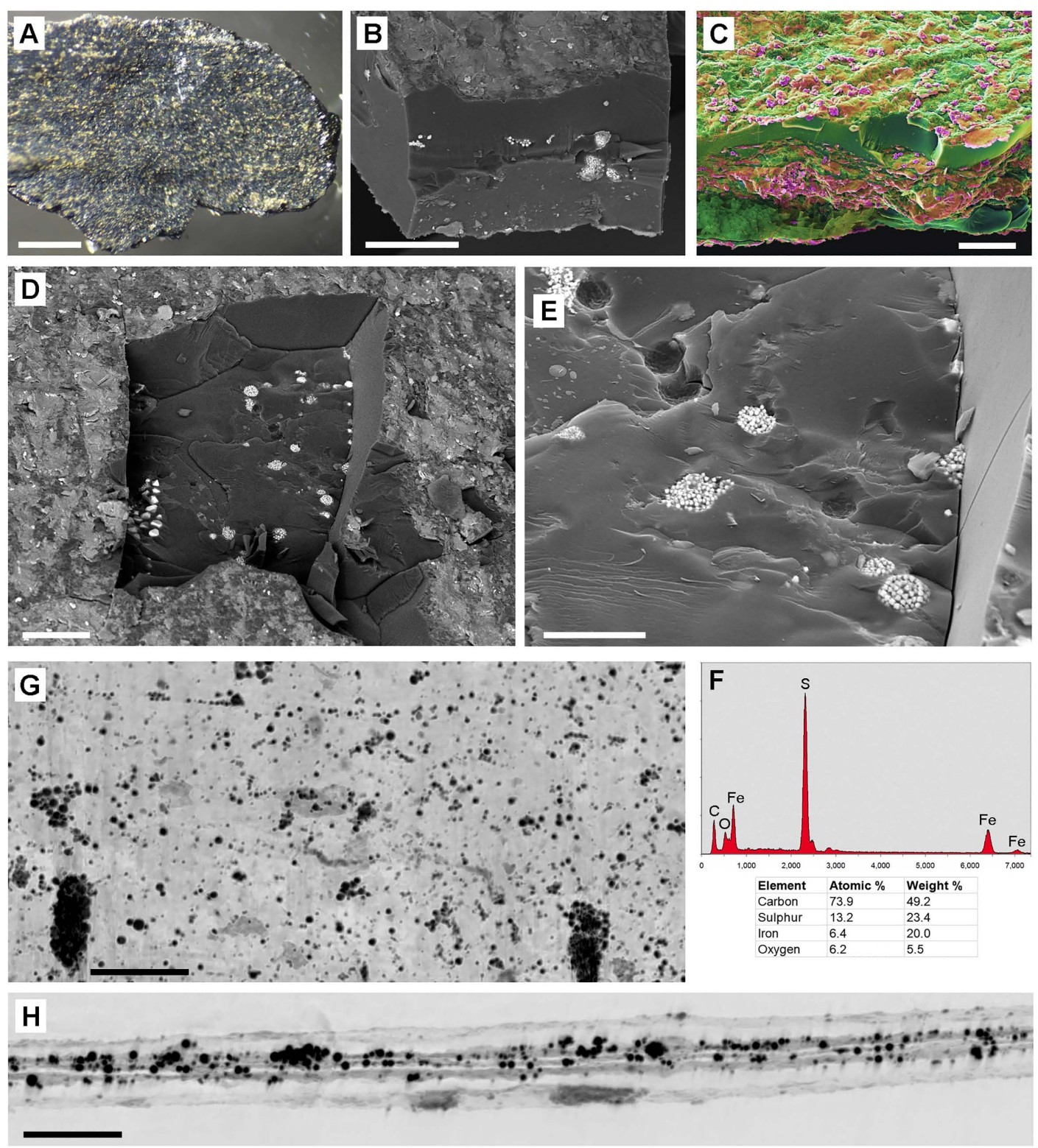

**Fig 5. Devonian fossil leaves assigned to *Cyclostigma kiltorkense* with granular inclusions** (A-H): Images of a fragment of carbonized leaf. (A): Overview by stereo-microscope shows the surface covered with small pyrite particles. (B-E): SEM images (compositional contrast, BSE). (B, C): Views on the edge of fragment show mineral inclusions in the middle of the 75 μm thick carbonized layer, either pure pyrite granules (B) or mixed with Si-Al

| Element | Atomic % | Weight % |
|---------|----------|----------|
| Carbon | 73.9 | 49.2 |
| Sulphur | 13.2 | 23.4 |
| Iron | 6.4 | 20.0 |
| Oxygen | 6.2 | 5.5 |

oxides (C). (D-E): Locations where the upper carbon layer is removed; pyrite crystals and framboids in the middle of the carbon layer are exposed. (F): EDX spectrum reveals composition of framboids (Fe, S). (G, H): µ-CT images of fragment. (G): plan view, entire thickness. (H): Reconstructed cross section shows Si-Al-mineral inclusions (gray) and pyrite granules (dark) in the middle of the fragment. The carbon layers appear transparent and almost invisible. Scale bars: (A) = 0.5 mm; (B, C, D) = 50 µm; (E) = 20 µm; (G, H) = 200 µm.

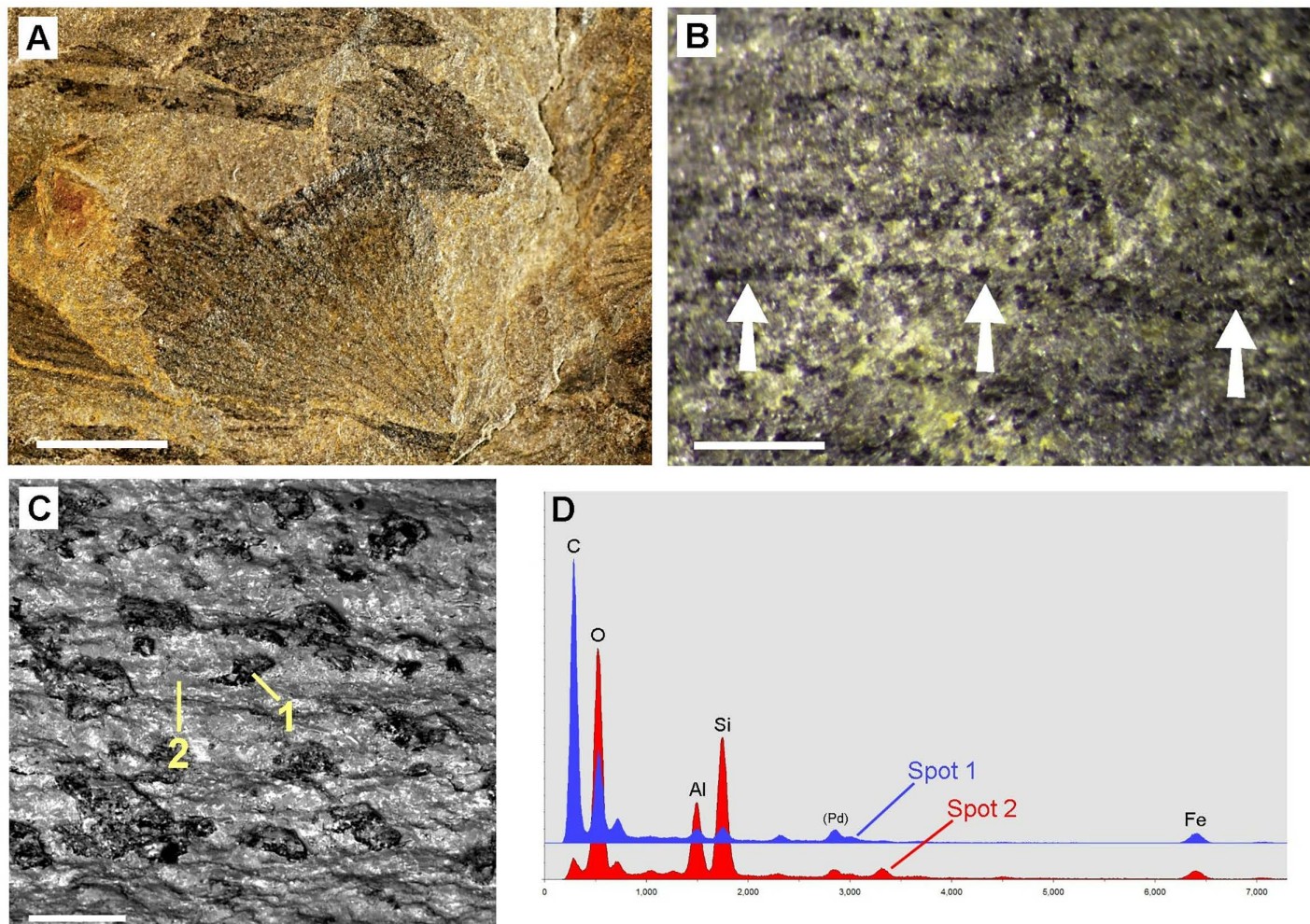

**Fig 6. Devonian fossil leaves assigned to *Archaeopteris roemeriana*** (A-D) (A): Overview photo. (B): LM image shows rows of black granules (arrows) below the removed carbon layer parallel to the venation. (C-D): SEM compositional contrast (BSE) image and element analyses of the black grains. Grains are mainly carbon (blue spectrum); the red spectrum shows composition of the sediment (oxides of Si, Al, Fe). Scale bars: (A) = 4 mm; (B) = 400 µm; (C) = 100 µm.

pale in contrast to the red or yellow background mineral and appeared darker in the BSE image (Fig 9 C). EDX element analyses of two spots marked in Fig 9 D revealed a composition of Si, Al, O (Si:Al ratio ca. 1:1) for the pale patches and Si, Al, O, and additional Fe in the sediment (Fig 9 E).

Seed ferns such as *Glossopteris* are extinct gymnosperms, and their closest extant relative plants are Cycadophyta (palm ferns, Cycads) [13]. For example, leaves of *Encephalartos lehmannii* (Cycadales, Zamiaceae) contain large CaOx

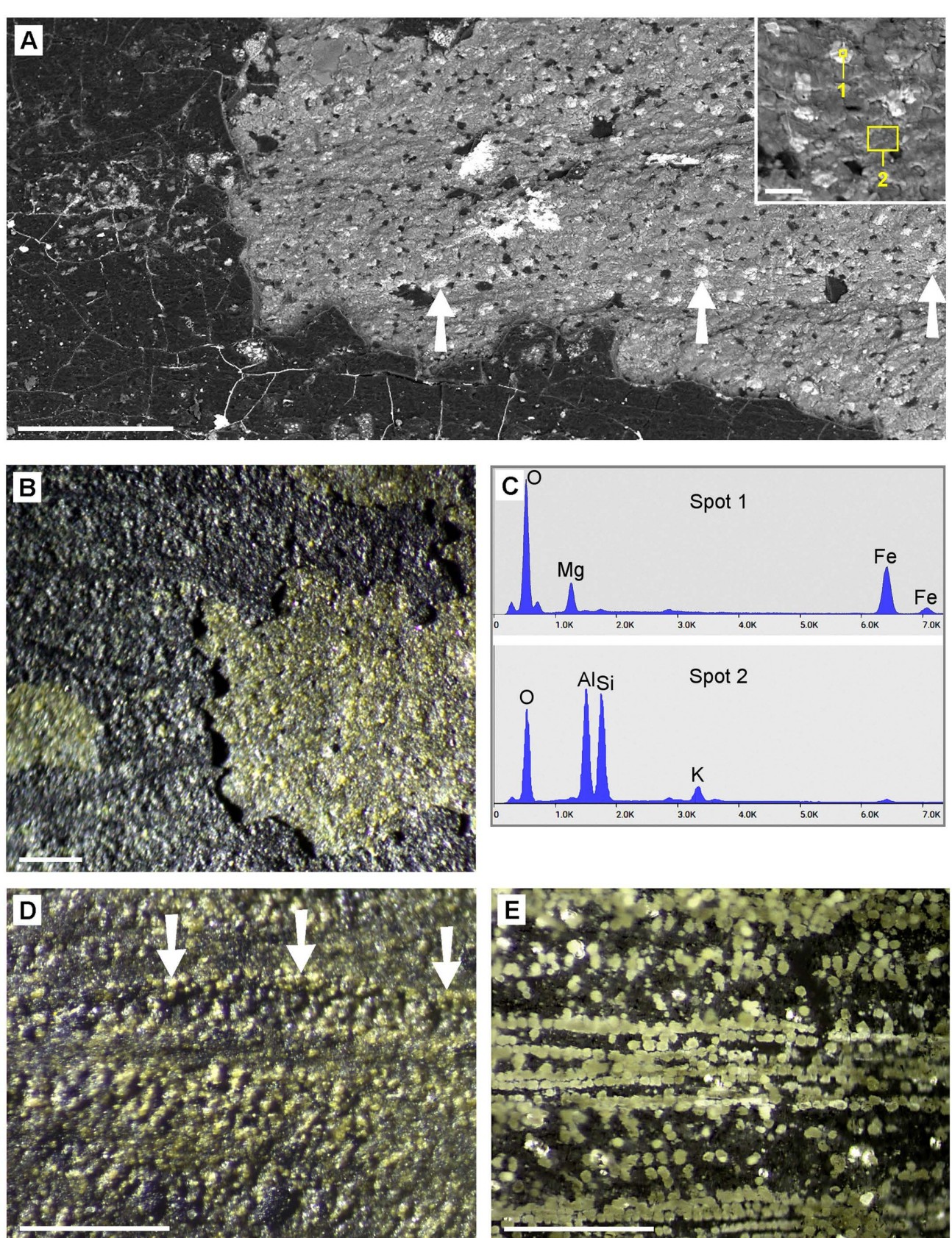

**Fig 7. Carboniferous fossil leaves:** (A-E): Fossil assigned to Paripteris gigantea in comparison with extant Encephalartos villosus leaf. (A): SEM BSE image shows a carbon layer (black), partially removed, and a pattern of bright granules (arrows) and dark carbon remnants on sediment under the carbon layer. Inset shows granules in detail. (B): LM image illustrates rough surface topography of carbon layer and yellow granules on sediment below carbon layer. (C): EDX spectra show compositions of structures in (A): small bright granules on sediment (Spot 1) are iron oxide (Fe, O) with magnesium; sediment (Spot 2, gray) consists of Si, Al, O. (D): Epi-illumination LM image of granular structure (arrows) under carbon layer in comparison with incinerated leaf of *Encephalartos villosus* (E), showing numerous CaOx druses in their natural distribution; same magnification. Scale bars: (A, B, D, E) = 400 µm; inset in (A) = 50 µm.

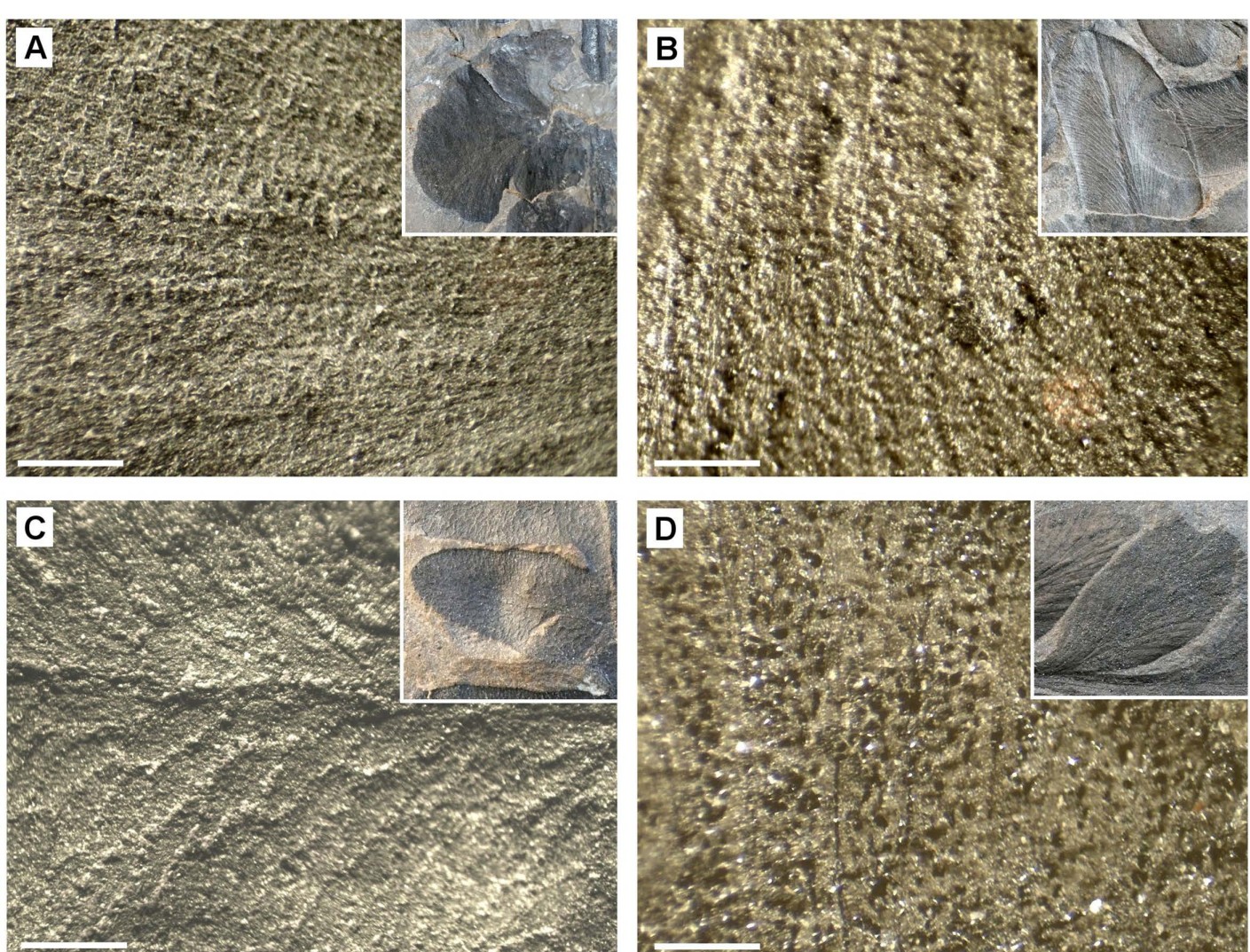

**Fig 8. Carboniferous fossil leaves from Upper Silesia Coal Basin, Poland.** LM images show granular patterns on the rough surface. (A): *Eusphenopteris striata*. (B): *Linopteris sp*. (C): *Lonchopteris rugosa*. (D): *Neuropteris rarinervis*. Insets show individual pinna. Scale bars: (A, B, D) = 200 µm; (C) = 500 µm.

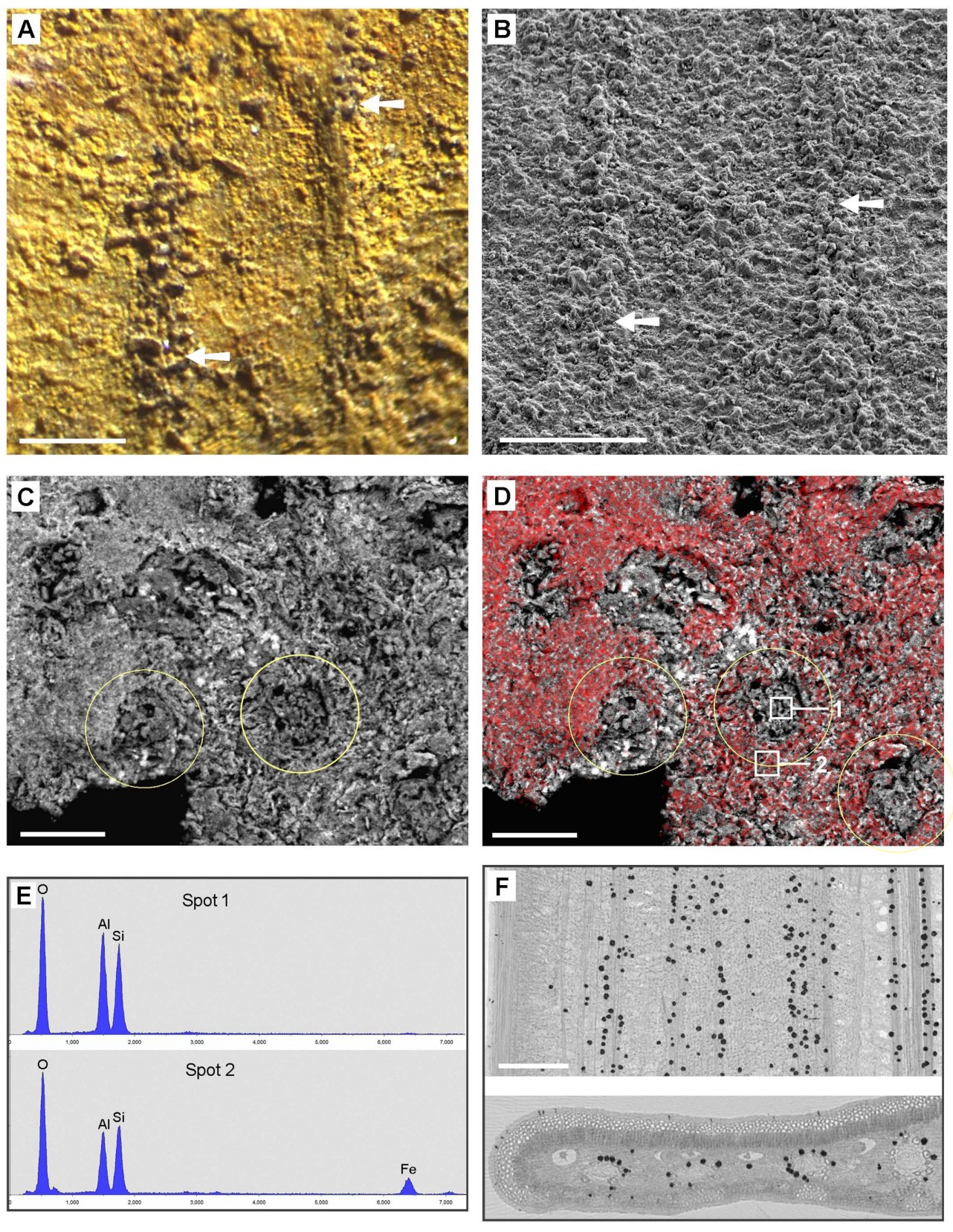

**Fig 9. Granular structures in the Permian mineralized fossil seed fern *Glossopteris sp.* in comparison with a fresh gymnosperm leaf.** (A-B) LM and SEM images of grainy parenchyma remnants along venation (arrows). (C-E) Granular structures in a flat fracture surface recognizable by their texture and slightly different composition. (C): BSE image; (D): BSE image with overlay of Fe element mapping image (Fe content = red). (E): EDX spectra of granules (indicating Si, Al, O) and sediment (indicating Si, Al, O, and Fe) of Spot 1 and 2 in (D). (F): µ-CT images of fresh leaf of *Encephalartos lehmannii* (Zamiaceae), a cycadophyte closely related to seed ferns, showing distribution of CaOx druses. Scale bars: (A, B, F) = 400 µm; (C, D) = 40 µm.

druses along the venation, and their distribution pattern resembles those of the granular structures in *Glossopteris*. (Fig 9 F; Fig 2 A). However, we found varying distribution patters within the same genus: in *E. lehmannii* druses were arranged along the venation; in *E. senticosus* they appeared randomly distributed (images not shown).

**Triassic.** *Otozamites brevifolius* (Fig 10 A-D) belongs to the Bennettitales, also known as cycadeoids. The samples were completely mineralized. Granular structures with sizes of 30–50 µm were visible by LM and in the SEM (Fig 10 A, B). EDX analyses showed similar composition for granules and surrounding material (Si, Al, O, little K and Fe; Fig 10 D). Thus, the granules could be recognized only by their morphology and texture.

**Tertiery, Eocene.** *Ginkgo adiantoides* samples are from Kamloops, British Columbia, Canada; Early Eocene, Tranquille. Formation of the Kamloops Group is descibed by Greenwood et al. (2016) [23].

An Early Eocene leaf of *Ginkgo adiantoides* (Fig 10 E-H) consisted of white mineralized remains of parenchymatic tissue. Striking white granular structures were clearly visible under the LM (Fig 10 F). Element analyses with SEM revealed that the white granules and the surrounding white layer consist of silica (Fig 10 G-H). The distribution of the granules in the parenchymatic tissue and resembles the patterns of druses in extant *Ginkgo biloba* or *Encephalartos lehmannii* leaves (Fig 2 A), indicating that they may be casts of druses. In parts of the leaf, we found also dark brown granules of similar size, which were not yet analyzed by EDX (arrows in Fig 10 F).

Table 2 summarizes the results of the analyses of the samples presented in this study.

## Discussion

Recognizing traces of CaOx crystals in fossil leaves is still in the beginning and our presentation is intended to draw the attention of paleontologists to structures that have received little attention to date. Even in the oldest investigated plants, from the Late Devonian, clear traces of CaOx druses are preserved. As already mentioned in the introduction, spherical structures that we interpret as casts of CaOx druses have often been observed but misinterpreted. The same seems to apply to framboids, which are often found in fossil plant remains [28,29]. One hypothesis suggests growth in sediment pores [28]. However, none of the models considered the spaces of the druses, after their decay or replacement, as sites of framboid formation. The fossil record of CaOx crystals in plants can provide information on the evolution of physiological and biochemical CaOx production as well as co-evolutionary aspects of CaOx crystals and druses that can have a mechanical role in defense.

### Evolution and functions of CaOx

The development of CaOx deposits in higher plants during their early evolution should be seen in the context of calcium metabolism in plants, particularly with the role of Ca for the stability of cell walls and entire plant bodies. Relatively large amounts of Ca occur in the cell walls in association with pectin and provide hardness and stability. Lower plants (ferns, mosses), compared to seed plants, contain lower concentrations of Ca and less common CaOx, only in form of small crystals [30]. The regular occurrence of CaOx in gymnosperms and angiosperms indicates an early begin of its evolutionary diversification. In modern gymnosperms, CaOx variety is still limited: conifers contain mostly small crystals attached to the cell walls, whereas cycads and Ginkgo are richly equipped with large druses, mainly along the venation [31]. Angiosperms have developed a much greater diversity of CaOx deposits than gymnosperms in both, morphology and distribution patterns. Druses and large prismatic crystals prevail in many dicotyledons. In leaves and bark of many trees, more than 90% of the total Ca content can be found as CaOx [32,33].

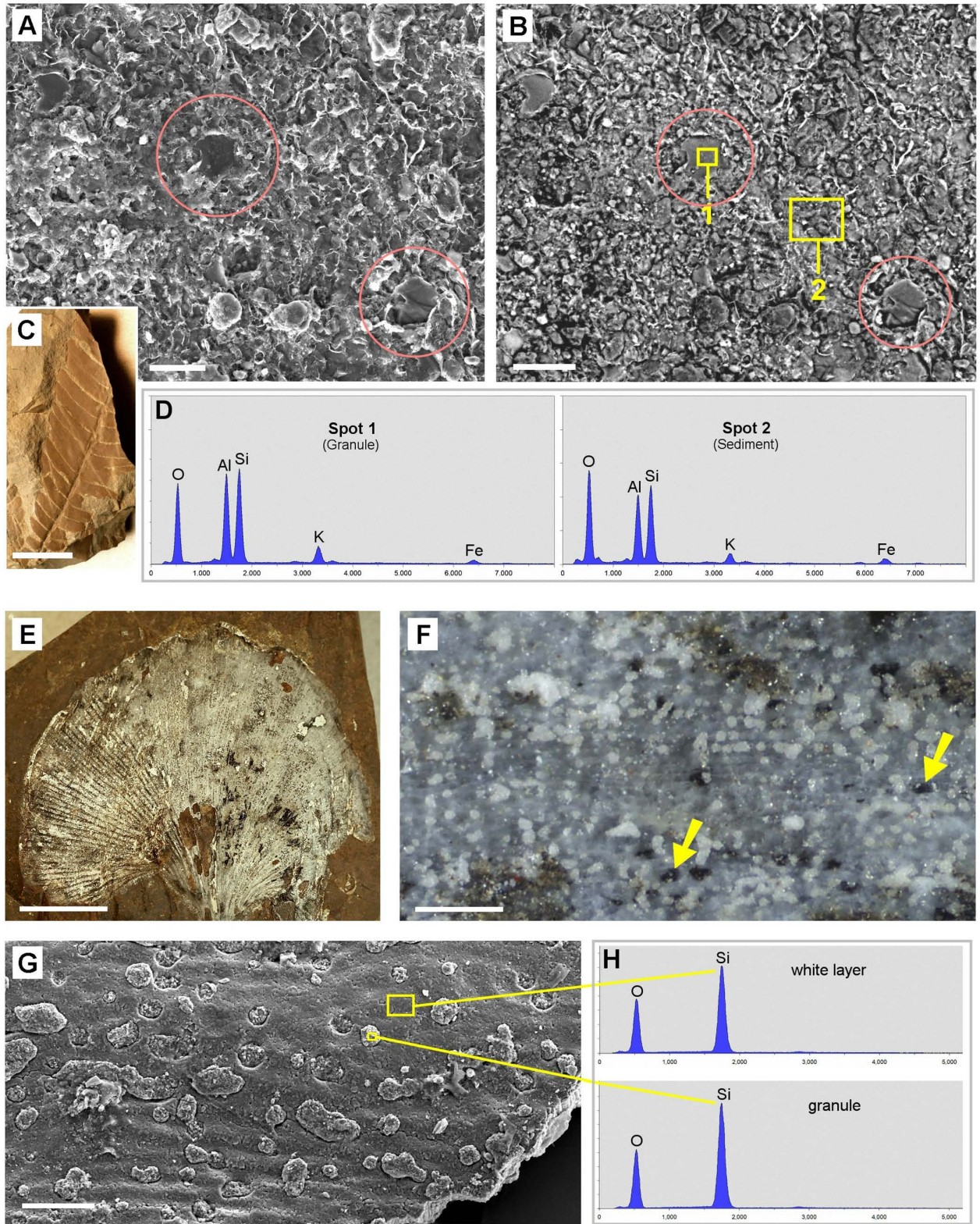

**Fig 10. Samples of Triassic and Tertiary leaf fossils.** (A-D): *Otozamites brevifolius* (Triassic). (A): Topography SEM image of granules. (B): Compositional contrast (BSE) image of the same area indicates homogeneous composition. Identical grains are encircled in both images. EDX analyses (D)

reveal similar composition (Si, Al, O, K, traces of Fe) for grains and sediment matrix. (C): Overview of the sample. (E-H): Fossil leaf of *Ginkgo adiantoides* with granular structures in remnants of parenchyma layer. (E, F): Photo of entire leaf and LM image showing venation and granules. Few dark brown granules are marked with arrows. (G, H): SEM image and EDX spectra of white layer and white granules indicating deposit as $SiO_2$. Scale bars: (A, G) = 50 µm; (B) = 30 µm; (C) = 20 mm; (E) = 10 mm; (F) = 200 µm.

**Table 2. Composition of granular structures and distribution patterns in fossil leaves. (n.a. = not analysed).**

| Fossil site; Age | Fossil species | Composition, Size of Granules, Distribution pattern | Composition of Matrix |
|---|---|---|---|
| Bear Island Late Devonian; 370 Ma | *Cyclostigma kiltorkense* (Fig 5) | Fe, S (Pyrite) 15 µm Mesophyll, randomly | |
| | *Archaeopteris roemeriana* (Fig 6) | C, traces Fe 60 µm along veins | Si, Al, O |
| Upper Silesian, Poland, Carboniferous, Westfalium „B"; 310 Ma | *Paripteris gigantea* (Fig 7) | Fe, O, Mg 15-30 µm along veins | Si, Al, O, traces K |
| Upper Silesia, Poland, Carboniferous, Westfalium B; 310 Ma | *Eusphenopteris striata* (Fig 8 A) | n.a. ca. 20 µm along veins | n.a. |
| | *Linopteris sp.* (Fig 8 B) | n.a. 15−0 µm along veins | n.a. |
| | *Lonchopteris rugosa* (Fig 8 C) | n.a. along veins | n.a. |
| | *Neuropteris rarinervis* (Fig 8 D) | n.a. 20-25 µm along veins | n.a. |
| Dunedoo, Australia, Upper Permian Ca. 275 Ma | *Glossopteris sp.* (Fig 9) | Si, Al, O 30-50 µm along veins | Si, Al, O, trace Fe |
| Unternschreez Bayreuth, Germany Triassic, Rhaetian; 204 Ma | *Otozamites brevifolius* (Fig 10 A-D) | Si, Al, O, trace K ca. 30 µm Mesophyll, randomly | Si, Al, O, trace K |
| Kamloops, Canada Tertiary, Early Eocene 50 Ma | *Ginkgo adiantoides* (Fig 10 E-H) | Si, O 20-40 µm along veins | Si, O |
| Eckfeld, Germany Eocene; 44.3 Ma | not identified (Fig 3) | Fe, S (Pyrite) 20-50 µm Mesophyll, randomly | Si, Al, O |
| Willershausen, Germany Pliocene; 3 Ma | *Betula* sp. (Fig 4 A, B) | Si, Al, Ca, S, O (incl. CaSO4) 20-60 µm | C, S |
| | *Liriodendron tulipifera* (Fig 4 C-E) | C, traces O, S 40 µm Mesophyll, randomly | C, traces O, S |

Remarkably, modern plant orders such as Poales (grasses) also contain only little amounts of Ca and rarely CaOx, but silica inclusions (phytoliths) are common [34].

Hypotheses on the function of CaOx biominerals are manifold. According to Franceschi and Nakata (2005) [7] there are a few most commonly mentioned functions: calcium regulation, plant protection/ defense, and detoxification of aluminum

(and other heavy metals). As a result of high Ca intake of the plants, CaOx deposits could serve as a Ca reserve or for the sequestration of excess Ca [1,35].

The high diversity of CaOx forms, particularly in angiosperms, indicates that functional developments are often isolated solutions in specific plant groups. For example, the stinging hairs of the plant *Tragia ramosa* have an apex that consists of an elongated stinging cell containing a large needle-shaped styloid CaOx crystal with a groove along one edge and a branched base. At contact, the toxin is injected under the skin, where it causes the dermal irritation or the perception of a stinging feeling [6,36,37]. Generally, the relevance of CaOx for defense should be treated with caution, since CaOx is a quite soft mineral. However, sharp-tipped druses and raphides in poisonous plants, e.g., many Araceae, may be effective by injuring the skin in the mouth of grazing animals, enabling the penetration of toxins.

The fossil record of CaOx crystals in plants can provide information on the evolution of physiological and biochemical CaOx production as well as co-evolutionary aspects of CaOx crystals and druses that can have a mechanical role in defense. In the oldest investigated plants, from the Late Devonian, clear traces of CaOx druses are preserved. Also, the as-yet oldest known feeding traces on plants derive from the Devonian [38]: Middle Devonian liverwort remains from the Catskill Delta deposit of eastern New York state, showing external foliage feeding and galling.

In general, the results presented here, don´t facilitate an explicit preference of a specific existing hypothesis on the possible function of CaOx. But they expand the fundamentals of the hypotheses and will allow a more precise focus of investigations in the future.

**Preservation quality.** The findings show that the preservation of former CaOx crystal casts is not limited to younger periods. Devonian plants can in some cases exhibit better preservation of CaOx remains than Tertiary fossils. Differences in fossilization conditions in various sites of diverse ages may be influential parameters. Therefore, we need more information about diagenetic and general taphonomic processes for a better understanding of the fate of CaOx crystals and druses during fossilization. A first attempt on the probable fossilization process has been represented in Malekhosseini et al. (2022) [8]. The chemical analysis by EDX clearly show that the composition of the inorganic elements in the refilled druses is variable. Mainly they include $SiO_2$ (Fig 10), Si-Al oxide (Fig 8), pyrite (Figs 3, 5), iron oxide (Fig 7), which were common in almost all of the sites but the substitution pattern of them in fossil leaves was different. In fossils of homogeneous material, the CaOx casts can be recognized by their granular or spherical shape, texture, and characteristic distribution. Many fossils consist of various materials, and in many cases the composition of the granular casts differs from that of the sediments (see Figs 3, 4, 5). This indicates that the CaOx casts may have been refilled at a different time than the diagenesis of the sediment. The minerals that fill the casts can also fill other spaces; pyrite deposits are known to occur in various sizes and locations. Silicified plants, in which the finest cell structures can often be recognized, are particularly promising for finding traces of druses or crystals.

The identification of refilled CaOx casts in leaves with thicker cuticle and mesophyll layer seems to be more reliable than in other leaves. CaOx druses are part of the leaf mesophyll, thus their traces in fossils are found in mesophyll remnants. They cannot be found in fossils which consist only of the cuticle.

The granular shape, texture, grain size, and composition of the sediment surrounding the CaOx casts are the main factors in recognizing the casts in fossil leaves in a particular sediment matrix [8]. However, even in the best-preserved fossils only a small fraction will show unambiguous traces of CaOx druses. In many species the original leaves contain only few large druses, others contain angular solitary crystals instead of druses. Their fossilized traces are harder to recognize and the discrimination from sediment contents is often unreliable. In many fossil samples, especially those of angiosperms, we also found abundant smaller granular structures that may be casts of CaOx crystals. However, their identification seemed too uncertain to us, because their distribution patterns were not distinctive enough. Advantages in pattern recognition and improved microscopy techniques, including micro-CT, will help to identify former biomineral structures in fossils. There is a rich fossil record and the reconstruction of the evolutionary history of one of the most important biomineral in plants is a promising field of research.

## Acknowledgments

We give special thanks to Dr. Cornelia Löhne for collecting plant material from the Bonn University Botanic Gardens. We would also like to give our special thanks to Dr. Ludwig Luthardt, curator of the (paleo-) botanical collection at the Museum für Naturkunde, Berlin. We thank Alexander Blanke and Dagmar Wenzel (Institute of Evolutionary Biology and Ecology, University of Bonn) for the support with X-ray and µ-CT images.

## Author contributions

**Conceptualization:** Mahdieh Malekhosseini, Victoria E. McCoy, Torsten Wappler, Jes Rust.

**Investigation:** Jes Rust.

**Methodology:** Mahdieh Malekhosseini, Hans-Jürgen Ensikat.

**Resources:** Mahdieh Malekhosseini, Torsten Wappler, Jes Rust.

**Supervision:** Jes Rust.

**Visualization:** Mahdieh Malekhosseini, Hans-Jürgen Ensikat, Victoria E. McCoy, Torsten Wappler, Jes Rust.

**Writing – original draft:** Mahdieh Malekhosseini, Hans-Jürgen Ensikat, Jes Rust.

**Writing – review & editing:** Victoria E. McCoy, Torsten Wappler, Jes Rust.

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
