## [Decision Letter · Decision Letter 0]

2 Apr 2025

Dear Dr. Malekhosseini,

Thank you for submitting your manuscript to PLOS ONE. After careful consideration, we feel that it has merit but does not fully meet PLOS ONE’s publication criteria as it currently stands. Therefore, we invite you to submit a revised version of the manuscript that addresses the points raised during the review process.

We look forward to receiving your revised manuscript.

Kind regards,

Borja Cascales-Miñana

Academic Editor

PLOS ONE

2. In your manuscript, please provide additional information regarding the specimens used in your study. Ensure that you have reported human remain specimen numbers and complete repository information, including museum name and geographic location.

For more information on PLOS ONE's requirements for paleontology and archeology research, see https://journals.plos.org/plosone/s/submission-guidelines#loc-paleontology-and-archaeology-research .

3. In the online submission form, you indicated that [The data that support the findings of this study are based on microscopic images which are archived in the Microscopy/SEM facilities of the Bonn Institute of Organismic Biology (BIOB), University of Bonn, Nussallee 8, 53115, Bonn, Germany. Images are available on request.

Plant collection statements

All plant samples collected in this study were taken from species cultivated in the Botanical Garden, Bonn. This sample collection complies with relevant institutional, national, and international guidelines and legislation.].

Additional Editor Comments:

Your paper although interesting and scientifically relevant presents a series of points that should be addressed before an eventual acceptance. Please, revise your MS according to received comments to prepare an improved version of your work.

Reviewers' comments:

Reviewer's Responses to Questions

**Comments to the Author**

1. Is the manuscript technically sound, and do the data support the conclusions?

Reviewer #1: Yes

Reviewer #2: Partly

2. Has the statistical analysis been performed appropriately and rigorously?

Reviewer #1: N/A

Reviewer #2: N/A

3. Have the authors made all data underlying the findings in their manuscript fully available?

Reviewer #1: Yes

Reviewer #2: Yes

4. Is the manuscript presented in an intelligible fashion and written in standard English?

Reviewer #1: Yes

Reviewer #2: Yes

Reviewer #1: This study extensively studies the granular structures in plant leaves of different geological ages and makes morphological and distributional comparisons with extant leaves to support the interpretation of CaOx druse remains. The figures are informative and the text has been improved from the last version. However, the logic of the CaOx druse interpretation still could be more quantitive, and a clearer conclusion and more discussions supporting the conclusion are required. So I suggest a major revision.

Major issues:

Abstract:

1. The Abstract is worth a better beginning sentence. The authors can consider to stress the physiological importance of CaOx in plants, and possibly their application for taxonomy, paleophysiology, or paleoenvironmental reconstruction. These would let readers know more about the importance of studying fossil CaOx.

2. At the end of the Abstract, the conclusion is still not clear. My comment in the last version was put here directly. I guessed that the authors want to conclude something like (1) “the early and extensive presence of CaOx in fossil plants” and (2) “the diverse chemical composition of former CaOx druses in fossils”. Whatever, the authors are expected to say out directly in the Abstract.

Introduction:

3. The references of previous studies are scattered over several paragraphs (Page 4 Line 100, Page 5 Line 113, Page 5 Line 120, Page 6 Line 136, Page 6 Line 145), please organize them together.

4. The sequence could be better. I suggest:

(1) previous studies;

(2) the previous studies are used to draw the interpretation bases of this study--“Traces of CaOx druses can be confidently identified due to their… (Page 6, line 139)”, “It has to be accentuated, that morphology, size, density and distribution pattern are verifiable characters of fossil CaOx druses (Page 5 Line 110)”;

(3) next begins with the motivation “Until now, traces of CaOx druses have only been…” (Page 5 Line 116);

(4) then what the authors did “In the present study we examined a variety of fossil leaves from a range of fossil sites from the Devonian to the Eocene… (Page 6, Line 132)”;

(5) at last, the questions to be answered (Page 7 Line 154).

Figures:

5. I found that Fig. 1G is a mirror of the insert of figure 1C in Malekhosseini et al. (2022). It is better to replace or remove Fig. 1G.

6. Where the enlarged figures were taken are not labelled from the fossils/leaves, this may raise questions about reliability of the figures.

Results:

7. The paragraph (Page 12 Line 265 to 283) about identification bases seems not part of the result. Consider moving to the Method section.

8. In each collection, morphological observations, EDX analyses, and interpretation are mixed. Consider making them into separate paragraphs. The discussion about the preservational processes can be put together in the Discussion section.

9. Generally, the logic of CaOx interpretation is not clear and therefore not convincing enough. Consider (1) making statistics of the granular size (with count numbers, mean values, and standard deviations) of extant and fossil specimens for direct comparison, (2) clarifying the distribution pattern and stressing the shared features with extant leaves.

Discussion:

10. Generally, the discussion about function has a weak connection with the observed results, especially the paragraph about CaOx raphides (Page 22 Line 514 to 523), which have not been reported here. I suggest the authors discuss the early emergence of CaOx druses, which may indicate that early plants have developed CaOx to defend feeding and to regulate calcium as in living plants.

11. Page 23 Line 547, the conclusion that ‘the composition … don’t follow any clear trends’ is not obvious enough to me. Consider adding a summarized table or analyses to support this conclusion. Alternatively, the authors could consider only concluding the “diverse” feature of the preservation.

Minor issues:

Abstract:

1. Page 2 Line 37, Page 6 Line 134, it could be better if not to use ‘traces of CaOx druses’ before the interpretation. Using ‘granular structures’ seems better.

2. Page 2 Line 43, “the relationship needs to be further summarized”, why not summarize it here?

Introduction:

3. Page 3 Lines 56 to 71, ‘different’ and ‘diverse’ are used many times here. I suggest the authors to summary the recognizable features (size, distribution, …) here instead of stressing the diversity, as they are the vital bases for the following interpretation as CaOx druses.

4. Page 3 Lines 65 to 67, this sentence could be moved forward to be before “the actual mechanisms…”.

5. Page 6 Lines 146 to 153, how did the ‘general scenario’ been developed before the results?

Figures:

6. ‘Angiosperms’ and ‘Dicotyledons’ (Page 4 Lines 82 to 83) need not be capitalized.

7. Page 4 Line 87, ‘to 80 μm’ may be ‘up to 80 μm’.

8. Page 4 Line 92, (K), how to distinguish CaCO3 from CaOx crystals?

9. Page 7 Lines 164 to 165, ‘the thick cell walls’ seems not related to the topic of this paper.

10. Page 7 Line 169, ‘regular druses leaf lamina’, here needs a preposition.

Methods:

11. Page 11 Line 228, the micro-CT method usually reports voltage, current, and voxel size, rather than proprieties of the sensor plate.

Results:

12. Page 13 Line 296, this sentence is repeated with Line 281.

13. Page 17 Line 397, ‘(Figs. 2A, 7E)’, but no “7E” is found.

14. Page 18 Line 415, ‘along the leaf venation’, it is hard to see in Fig. 6A. If hard to improve the tomographic signal, consider labelling the venation as in Fig. 6D.

15. Page 19 Line 449, it is confusing what ‘colourless’ refers to.

16. Page 20 Line 473, ‘visible by LM’, but no LM images in Fig. 9 A, B.

Discussion:

17. Page 23 Line 554, ‘quite early or much later before or after’ is a confusing phrase.

Other issues:

The grammar and spelling of the text need to be checked.

Reviewer #2: This manuscript presents a description of various putative calcium oxalate druses, in fossils

from the Devonian to the Neogene. Generally, this work is interesting but a clearer statement

of the goal and a summary of the results have to be done, to improve the paper.

Major comments:

1- Title and goal

“The evolutionary history‟ mentioned in the title is not reflected by discussion in the paper. I

would recommend changing the title to “the temporal/ historical distribution‟ rather than the

“evolutionary history‟. This goes along better with the general motivation of the study, which

appears to be the “identification and correct interpretation of CaOx structures‟, according to

an authors‟ response to reviewer 1. This aim must be stated more clearly, in addition to the

questions raised by the authors.

2- Accordance data - conclusions

The features used to recognize CaOX druses are varying across the paper: shape distribution

texture p 23 vs. granular shape, texture, grain size, composition of sediment surrounding p 24

vs. morphology and distribution p12 vs. distribution and texture later in the same page. This

lack of consistency clearly impairs the paper. Please clearly state them in the introduction or

at the beginning of the results.

A table summarizing the results obtained from all your fossils with the selected diagnostic

features will certainly help supporting your hypothesis and increase the clarity of the paper. A

new paragraph in the discussion, highlighting your arguments of interpretation and ruling out

other possibilities (see later paragraph about framboidal pyrite) will also be of critical use for

the audience, to interpret correctly comparable structures in other fossils.

3- Previous reviews

It is troubling that part of previous remarks by reviewers was addressed only partially or not

at all, without explanation. Please see further some of them that, in my point of view, were not

answered but must be addressed before publication.

Minor comment:

- Figures 1 and 2 are mentioned and explained in both the introduction and results.

Please only do it once.

- Figures are not cited (8C) or not in order (8F before 8E in legend)

- The energy range of all your EDS spectra is either blurred or truncated. In Figure 2I

the annotations added are also blurred. Please enhance the quality of such features.

- Results are intertwined with interpretation. Since you chose to have two separate

sections, please remove the latter from the results section (for instance: l.312, l.316,

l.370, l. 487

- Use of contractions and incomplete sentences must be avoided.

- They are redundancies :

compositions of CaOx cast varying; l.42 and 45

“leaves of many species, particularly dicotyledons, CaOx crystals‟ l.55 and 58;

CaOx forms l.56 and 68

“has been substituted […] surrounding sediment‟ l.312-313 and 316-317

- Back and forth within a paragraph of a same notion, with sometimes contradictory

data: “all parts‟ l.54 then “mainly intra-vacuolar membrane‟ l.66; “60-100μm diameter‟

l.280 vs. “30-80μm‟ l.292

- Typos: remove capital letter after “:‟ : l.147, l. 307

Page by page review:

l.180: Delete sentence, it is already stated Table 1.

l. 222: “the tescan SEM has low-vacuum capacity‟. More importantly, has it been used?

Otherwise, delete.

l.308 + legend Figure 3 : these are “framboids” not “granules‟. The term is very specific and

you should use it. See Rickard, 2021. Framboids.

https://doi.org/10.1093/oso/9780190080112.001.0001

l.326: if the scale is about 100μm, the structures are either 40 or 70 μm… Please check.

l.355: Pseudobornia ursina not mentioned Table 1

l.375: no analyses given to evidence the Si-Al nature of the sediment

l.474: iron is not mentioned while visible on EDS spectrum. Same l. 495.

l.549: CaSO4 has never been mentioned before. Can you explain?

Comments on previous reviews:

Reviewer 1 + Reviewer 2

R1 : 15. Page 16, Line 19-21. “This formation…Røedvika Formation (Horn & Orvin, 1928)”.

The stratigraphy study's history is irrelevant to the specimen's age and the paper's topic.

Not addressed.

R2 : - Page 16 it is strange to have a description of the stratigraphy of Bear Island deposit but

not of the other deposits

The large collection of Bear Island fossils is located in our institute in Bonn, thus all the data

are available.

But it is still irrelevant to the paper. Please delete.

R1 : 15. Page 29, Lines 6-7. "Differences in fossilization conditions in various sites". This

aspect of information was not shown in the paper. You can consider adding this information

to a summarizing table.

the most influencial and limiting“ deleted; so the statement seems to be correct

R2 :- Page 9: “The material derived from several institutes and museums‟ => in Table 1 they

all appear as coming from the Goldfuß-Museum, University of Bonn. They are no mentions

of the collections from Mainz, Göttingen… latter mentioned in the results. Please update the

Table. Please also include a column specifying the depositional conditions for each fossil site

(are they consistent?) and another for gymno/angiosperms (since you present the results

through that property; + monocotyledons vs. dicotyledones if angiosperms).

Ok; missing data are included in Table 1

The last sentence has not been addressed. Since you mentioned fossilization conditions in

your discussion, the role of depositional conditions could be further extended, and references

provided.

Reviewer 1

14. The "Fossil record of CaOx cast preservation" section does not provide much more

information than the Results section. You can consider making a table to summarize these

granular structures' distribution, morphology, and chemistry from different species, ages, and

localities.

Ok; we have deleted most of the redundant information and re-arranged the Discussion.

The table asked would be very useful.

38. Page 29, Line 16. You can show the "identical or different" compositions of the refilled

druses and the sediments in the form of a table.

Seems not helpful, due to high variability.

Which is the very reason why a table will help gain in clarity !

13. Page 14, Line 7. "Iron oxide caused by storage in glycerol" needs a reference.

Additional information or images on these contamination seem unnecessary and confusing.

I do not agree with that. Glycerol is supposed to store specimens away from the atmosphere,

thus avoiding oxidation by dioxygen or humidity. To me, it is quite surprising that storage in

glycerol causes oxidation. So please delete or add a reference as requested.

Reviewer 2

Major comments

1- Diagnostic characters of CaOx druses lately replaced; […]

(i) from the introduction, it comes out that form, size, density and locations of CaOx druses

may vary greatly from one leave to another […]

Explanation: Indeed, CaOx occurs in variable forms such as large and small crystals,

spherical druses, needles and so on. Fortunately, relatively large druses occur typically in

gymnosperms (except most conifers), such as Cycadophyta, and in many dicotyledones. This

spherical shape and the distribution patterns, e.g., along the veins, are the basic factors for the

recognition. However, it requires the knowledge of CaOx distribution patterns in extant

plants. Identification of casts from different CaOx forms such as crystals or raphides would

require excellently preserved fossils, and the use of most-suitable examination techniques,

without the usual restrictions for museum collections.

Maybe a clear sentence stating that (i) you compare with the actual record and (ii) you

describe specific distribution/ shape/ size/ textures for each type of plants will help. (see

previous major comment 2) of this review.

(iii) You indicate (page14) that CaOx druses are substituted by pyrite. However, framboidal

pyrite can grow on leaves without being a replacement of CaOx druses: see Rickard, D.,

Grimes, S., Butler, I., Oldroyd, A., & Davies, K. L. (2007). Botanical constraints on pyrite

formation. Chemical Geology, 236(3-4), 228-246. So what are your points to claim that they

are characteristics of former CaOx druses, and not the result of a simple precipitation onto the

surface?

The occurrence within the fossil leaf, in the remnants of the parenchyma, is the main

indication for its origin from a spherical bio-structure.

The authors of the paper mentioned above did obtain pyrite crystals within the parenchyma,

which proves that such crystals can be the direct result of precipitation and not due to the

substitution of former CaOx druses as you state. Please bring other points to strengthen your

hypothesis here, or remove these data.

2- Chemical analysis;

For fossil leaves, can you please provide systematically an analysis of the structures and of

the hosting rock? For instance, you state for Figure 3F that the dark appearance of the

granules and their EDS spectrum indicate an organic deposition, but the matrix just above is

also really dark. Does it contain carbon? Finally, most of the EDS spectra lack elemental

attribution to peaks. Please ensure that they are all labelled.

Element spectrum peaks have been labelled.

The remark has only been partially addressed. Could you please provide EDX spectra of the

sediment for Fig 3D, 3F and 4? Since you have stated the composition of sediment

surrounding as a factor of identification of CaOX druses, this data must be consistently

provided.

Page-by-page comments

- Page 4, line 2: carbonates are among the few main minerals involved in preservation of

organisms (with phosphate, silica, …). How can CaCO3 be instable while allowing

exceptional preservation? You need a reference here, or explain further.

Ok; specified fossilization of plants“. As mentioned, most of our leaf fossil samples did not

contain any CaCO3 or CaOx; perhaps due to acidic environment during fossilization or decay.

This information is interesting and must be discussed and supported by data or references of

the depositional environment of your fossil deposits (in Table 1 for instance), as asked

previously).

- Page 10: how many leaves of each specie did you study? How were the leaves stored until

analysis? Is there any chance that pollution (I suppose Bonn is polluted as most cities in

Western Europe) may affect CaOx druses formation/ extant?

We studied leaves from >200 species in order to get an idea of the distribution patterns of

druses and crystals, but not for statistics. When the microscope images were consistent with

the overall results, then the images of a single leaf could be sufficient. Many species were

studied multiple times with different methods. However, the images show examples of druse

distributions, but not complete statistics on concentration and size.

This is very important as it enhances the representativeness of your data. It should be

mentioned in the material and methods part.

**Do you want your identity to be public for this peer review?** For information about this choice, including consent withdrawal, please see our Privacy Policy

Reviewer #1: No

Reviewer #2: No

---

## [Author Response · Author response to Decision Letter 1]

20 May 2025

Response to Reviewers

(Answers and explanations of the authors are in green color)

PONE-D-25-01413

The evolutionary history and fossil record of calcium oxalate druses in fossil leaves of angiosperms and gymnosperms.

PLOS ONE

Dear Dr. Malekhosseini,

Thank you for submitting your manuscript to PLOS ONE. After careful consideration, we feel that it has merit but does not fully meet PLOS ONE’s publication criteria as it currently stands. Therefore, we invite you to submit a revised version of the manuscript that addresses the points raised during the review process.

We look forward to receiving your revised manuscript.

Kind regards,

Borja Cascales-Miñana

Academic Editor

PLOS ONE

2. In your manuscript, please provide additional information regarding the specimens used in your study. Ensure that you have reported human remain specimen numbers and complete repository information, including museum name and geographic location.

For more information on PLOS ONE's requirements for paleontology and archeology research, see https://journals.plos.org/plosone/s/submission-guidelines#loc-paleontology-and-archaeology-research.

3. In the online submission form, you indicated that [The data that support the findings of this study are based on microscopic images which are archived in the Microscopy/SEM facilities of the Bonn Institute of Organismic Biology (BIOB), University of Bonn, Nussallee 8, 53115, Bonn, Germany. Images are available on request.

Plant collection statements

All plant samples collected in this study were taken from species cultivated in the Botanical Garden, Bonn. This sample collection complies with relevant institutional, national, and international guidelines and legislation.].

Additional Editor Comments:

Your paper although interesting and scientifically relevant presents a series of points that should be addressed before an eventual acceptance. Please, revise your MS according to received comments to prepare an improved version of your work.

Reviewers' comments:

Reviewer's Responses to Questions

Comments to the Author

1. Is the manuscript technically sound, and do the data support the conclusions?

Reviewer #1: Yes

Reviewer #2: Partly

2. Has the statistical analysis been performed appropriately and rigorously?

Reviewer #1: N/A

Reviewer #2: N/A

3. Have the authors made all data underlying the findings in their manuscript fully available?

Reviewer #1: Yes

Reviewer #2: Yes

4. Is the manuscript presented in an intelligible fashion and written in standard English?

Reviewer #1: Yes

Reviewer #2: Yes

5. Review Comments to the Author

Reviewer #1: This study extensively studies the granular structures in plant leaves of different geological ages and makes morphological and distributional comparisons with extant leaves to support the interpretation of CaOx druse remains. The figures are informative and the text has been improved from the last version. However, the logic of the CaOx druse interpretation still could be more quantitive, and a clearer conclusion and more discussions supporting the conclusion are required. So I suggest a major revision.

Major issues:

Abstract:

1. The Abstract is worth a better beginning sentence. The authors can consider to stress the physiological importance of CaOx in plants, and possibly their application for taxonomy, paleophysiology, or paleoenvironmental reconstruction. These would let readers know more about the importance of studying fossil CaOx.

Answer: We have added more details about biominerals and CaOx here and in the Introduction.

2. At the end of the Abstract, the conclusion is still not clear. My comment in the last version was put here directly. I guessed that the authors want to conclude something like (1) “the early and extensive presence of CaOx in fossil plants” and (2) “the diverse chemical composition of former CaOx druses in fossils”. Whatever, the authors are expected to say out directly in the Abstract.

Answer: We have set the focus more on the recognition of biomineral (CaOx) traces.

Introduction:

3. The references of previous studies are scattered over several paragraphs (Page 4 Line 100, Page 5 Line 113, Page 5 Line 120, Page 6 Line 136, Page 6 Line 145), please organize them together.

4. The sequence could be better. I suggest:

(1) previous studies;

(2) the previous studies are used to draw the interpretation bases of this study--“Traces of CaOx druses can be confidently identified due to their… (Page 6, line 139)”, “It has to be accentuated, that morphology, size, density and distribution pattern are verifiable characters of fossil CaOx druses (Page 5 Line 110)”;

(3) next begins with the motivation “Until now, traces of CaOx druses have only been…” (Page 5 Line 116);

(4) then what the authors did “In the present study we examined a variety of fossil leaves from a range of fossil sites from the Devonian to the Eocene… (Page 6, Line 132)”;

(5) at last, the questions to be answered (Page 7 Line 154).

Answer: Thanks for the valuable suggestions. We re-arranged the Introduction accordingly.

Figures:

5. I found that Fig. 1G is a mirror of the insert of figure 1C in Malekhosseini et al. (2022). It is better to replace or remove Fig. 1G.

Answer: We have replaced Fig. 1G.

6. Where the enlarged figures were taken are not labelled from the fossils/leaves, this may raise questions about reliability of the figures.

Results:

7. The paragraph (Page 12 Line 265 to 270) about identification bases seems not part of the result. Consider moving to the Method section.

Answer: Ok, done.

8. In each collection, morphological observations, EDX analyses, and interpretation are mixed. Consider making them into separate paragraphs. The discussion about the preservational processes can be put together in the Discussion section.

9. Generally, the logic of CaOx interpretation is not clear and therefore not convincing enough. Consider (1) making statistics of the granular size (with count numbers, mean values, and standard deviations) of extant and fossil specimens for direct comparison, (2) clarifying the distribution pattern and stressing the shared features with extant leaves.

Answer: We have included far more details about the granular structures, their sizes, composition and distribution, in the Results and additional Table 2.

Discussion:

10. Generally, the discussion about function has a weak connection with the observed results, especially the paragraph about CaOx raphides (Page 22 Line 514 to 523), which have not been reported here. I suggest the authors discuss the early emergence of CaOx druses, which may indicate that early plants have developed CaOx to defend feeding and to regulate calcium as in living plants.

Answer: Yes; we made substantial changes in the Discussion according to these suggestions.

11. Page 23 Line 547, the conclusion that ‘the composition … don’t follow any clear trends’ is not obvious enough to me. Consider adding a summarized table or analyses to support this conclusion. Alternatively, the authors could consider only concluding the “diverse” feature of the preservation.

Answer: Indeed, the composition of the casts appears as variable as the composition of the surrounding sediments. We have written this statement more clearly.

Minor issues:

Abstract:

1. Page 2 Line 37, Page 6 Line 134, it could be better if not to use ‘traces of CaOx druses’ before the interpretation. Using ‘granular structures’ seems better.

Answer: Ok; done.

2. Page 2 Line 43, “the relationship needs to be further summarized”, why not summarize it here?

Answer: ok; „summarized“ replaced by „investigated“.

Introduction:

3. Page 3 Lines 56 to 71, ‘different’ and ‘diverse’ are used many times here. I suggest the authors to summary the recognizable features (size, distribution, …) here instead of stressing the diversity, as they are the vital bases for the following interpretation as CaOx druses.

Answer: Yes, we have added more details in the Introduction.

4. Page 3 Lines 65 to 67, this sentence could be moved forward to be before “the actual mechanisms…”.

Answer: Ok; done.

5. Page 6 Lines 146 to 153, how did the ‘general scenario’ been developed before the results?

Answer: It is a scenario (hypothesis) introduced in a previous publication. We have rewritten this sentence more precisely.

Figures:

6. ‘Angiosperms’ and ‘Dicotyledons’ (Page 4 Lines 82 to 83) need not be capitalized.

7. Page 4 Line 87, ‘to 80 μm’ may be ‘up to 80 μm’.

Answer: Points 6 and 7: done.

8. Page 4 Line 92, (K), how to distinguish CaCO3 from CaOx crystals?

Answer: The identity of the crystals has been verified multiple times with different methods, including Raman spectroscopy, solubility in weak acids, by us and other researchers.

9. Page 7 Lines 164 to 165, ‘the thick cell walls’ seems not related to the topic of this paper.

Answer: We think, it is worth to note that thick cell walls in certain leaves may be the cause for fossil leaves that consist of relatively thick carbon layers.

10. Page 7 Line 169, ‘regular druses leaf lamina’, here needs a preposition.

Answer: Ok; done.

Methods:

11. Page 11 Line 228, the micro-CT method usually reports voltage, current, and voxel size, rather than proprieties of the sensor plate.

Answer: Ok; done.

Results:

12. Page 13 Line 296, this sentence is repeated with Line 281.

Answer: This sentence and images of conifers deleted.

13. Page 17 Line 397, ‘(Figs. 2A, 7E)’, but no “7E” is found.

Answer: „7E“ replaced by „6E“.

14. Page 18 Line 415, ‘along the leaf venation’, it is hard to see in Fig. 6A. If hard to improve the tomographic signal, consider labelling the venation as in Fig. 6D.

Answer: Arrows included.

15. Page 19 Line 449, it is confusing what ‘colourless’ refers to.

Answer: colorless replaced by pale.

16. Page 20 Line 473, ‘visible by LM’, but no LM images in Fig. 9 A, B.

Answer: we preferred the SEM images due to better clarity.

Discussion:

17. Page 23 Line 554, ‘quite early or much later before or after’ is a confusing phrase.

Answer: Ok; sentence re-written.

Other issues:

The grammar and spelling of the text need to be checked.

Reviewer #2: This manuscript presents a description of various putative calcium oxalate druses, in fossils

from the Devonian to the Neogene. Generally, this work is interesting but a clearer statement

of the goal and a summary of the results have to be done, to improve the paper.

Major comments:

1- Title and goal

„The evolutionary history‟ mentioned in the title is not reflected by discussion in the paper. I

would recommend changing the title to „the temporal/ historical distribution‟ rather than the

„evolutionary history‟. This goes along better with the general motivation of the study, which

appears to be the „identification and correct interpretation of CaOx structures‟, according to

an authors‟ response to reviewer 1. This aim must be stated more clearly, in addition to the

questions raised by the authors.

Answer: We have included more information about plant systematic and evolutionary aspects in Introduction and particularly Discussio

---

## [Decision Letter · Decision Letter 1]

24 Jul 2025

Dear Dr. Malekhosseini,

Thank you for submitting your manuscript to PLOS ONE. After careful consideration, we feel that it has merit but does not fully meet PLOS ONE’s publication criteria as it currently stands. Therefore, we invite you to submit a revised version of the manuscript that addresses the points raised during the review process.

We look forward to receiving your revised manuscript.

Kind regards,

Borja Cascales-Miñana

Academic Editor

PLOS ONE

Journal Requirements:

Additional Editor Comments:

Please, follow the reviewer comments and correct the commented points.

Reviewers' comments:

Reviewer's Responses to Questions

**Comments to the Author**

Reviewer #1: (No Response)

Reviewer #2: (No Response)

2. Is the manuscript technically sound, and do the data support the conclusions?

Reviewer #1: Yes

Reviewer #2: Yes

3. Has the statistical analysis been performed appropriately and rigorously?

Reviewer #1: N/A

Reviewer #2: N/A

4. Have the authors made all data underlying the findings in their manuscript fully available?

Reviewer #1: Yes

Reviewer #2: Yes

5. Is the manuscript presented in an intelligible fashion and written in standard English?

Reviewer #1: Yes

Reviewer #2: Yes

Reviewer #1: The authors significantly improved the abstract and discussion sections and make a good table to summarise the abundant results observed, according to our suggestions. I found the paper generally meets the publication criteria but there are still some disorganisations in the text. So I suggest acceptance after a minor revision.

Here are some major points:

1. I agree with the Reviewer 2 that the title needs some changes. The ‘evolutionary history’ is not stressed in the paper at all. I have to stress that if you have a title like “the evolutionary history of calcium oxalate druses”, we will expect the reconstruction of the morphological or distributional changes of the druses along the phylogenetic tree.

2. As for the newly added Table 2, the authors only listed the sizes. (1) I do not understand why the authors do not make a statistic of the granule sizes, which would significantly help the comparison of sizes; (2) I suggest adding the morphology and distribution patterns to the table as well, which would help to support the interpretation as CaOx druse remains.

3. Related to the question (3), a discussion about the variation of composition inside each clade is needed.

4. In the discussion part, I want to see some words to exclude other possible interpretations of the granular structures, e.g. cell nuclei or special mesophyll cells.

Some minor points are here:

ABSTRACT:

1. Line 46, “These granular structures resembled patterns”, specify what pattern please.

2. Line 47, “can be interpreted”, you can directly say “here interpreted”.

3. Line 52, “the correlations still need to be investigated” can be deleted.

INTRODUCTION:

4. Line 62, here the proportion of angiosperms is listed, what about gymnosperms?

5. Line 109, “the obvious instability” necessarily needs references or detailed explanation!

6. Lines 118 – 121, I cannot understand why these patterns are “confident” and “verifiable” identifiers, give references please.

7. Lines 158 – 160, the sentence “In leaves of …” should not be put here but within the introduction part of living gymnosperm CaOx.

DISCUSSION:

8. Line 571, “frequently” should be replaced by a certain proportion from your results.

9. Line 579, need references or images for support.

10. Lines 580–581, this description of a previous study should be moved to the introduction part.

11. There are still some spelling mistakes in the text, please check them.

Reviewer #2: Compared to the previous version of the manuscript, the addition of a table and several changes in the figures have improved the quality of the data, enhancing the relation between data and conclusions. The abstract and introduction are also more informative but to my points of views, there are still some points that need to be addressed.

Major issues:

General:

1. For me, the revised manuscript and the revised manuscript with track changes are different! Particularly, the results paragraphs on Eckfeld and Willerhausen collections differ. In my opinion, the ‘revised’ version seems more completed.

2. The goal of the authors is much clearer and the work recentered on the recognition of CaOx druses. However, a hook to understand their importance is still missing (for instance in the abstract) and impedes to figure out the significance of your work. It’s unfortunate.

3. The authors state that pyrite framboids are substitutes of former CaOx druses. This statement is interesting but remains hypothetical: the size of pyrite framboids can go up to several hundreds of micrometers, contrary to the size of granules observed here in the fossil leaves (60 µm max. according to table 2). Several of the sentences would benefit from a more precautious approach (L321-324 of ‘revised version’ – not in track changes one + L524-525).

Material:

A response to a former question about the number of samples and specimens observed was directly added at the end of the material paragraph. It seems a bit clumsy and should be rephrased.

Results:

Consider moving the paragraph from l.280 to 291 to the method section (as it has been done for a former sentence), since it deals about the methodology of identification. Like this, and although it allows to understand your thought process, it is a bit hard to read, between two pieces of results.

The authors stress that in fossil leaves, ‘there was a high degree of uncertainty in the morphological interpretation’ (l281) while they latter state that ‘spherical morphology of suitable size’ (l294) is among the characteristic trends to identify Caox druses. Perhaps a consensus approach would be to state that in well-preserved fossils, morphology + distribution are the key characteristics but with poor preserved fossils, only the distribution was used?

Discussion:

I recommend moving the second sentence later in the text (after ref. 19 and 20) so the occurrence of druses in old fossil plants stresses their importance in providing information on evolution, etc …

Minor issues

- Typos: ‘whithout’, ‘staightforward’

- L42, compared with biomineral patterns in leaves -> compared with biomineral patterns in extant leaves?

- L48, depends on varying fossilization conditions -> depends on fossilization conditions?

- L178, leaf fossils -> fossil leaves?

- L458, ‘colour’ has been replaced by ‘composition’. But in the following, you’re definitively talking about colour since you mention ‘pale’ or ‘darker’ patches. Why not ‘by colour and composition (and texture)’?

**Do you want your identity to be public for this peer review?** For information about this choice, including consent withdrawal, please see our Privacy Policy

Reviewer #1: **Yes: ** Dr. Qingyi Tian

Reviewer #2: No

---

## [Author Response · Author response to Decision Letter 2]

12 Aug 2025

Here are some major points:

1. I agree with the Reviewer 2 that the title needs some changes. The ‘evolutionary history’ is not stressed in the paper at all. I have to stress that if you have a title like “the evolutionary history of calcium oxalate druses”, we will expect the reconstruction of the morphological or distributional changes of the druses along the phylogenetic tree.

Answer: We changed the title according the view of both reviewers.

Old title: The evolutionary history and fossil record of calcium oxalate druses in fossil leaves of angiosperms and gymnosperms.

New title:

Detection of traces of calcium oxalate druses in fossil leaves of angiosperms and gymnosperms from different sites and geological periods.

2. As for the newly added Table 2, the authors only listed the sizes. (1) I do not understand why the authors do not make a statistic of the granule sizes, which would significantly help the comparison of sizes; (2) I suggest adding the morphology and distribution patterns to the table as well, which would help to support the interpretation as CaOx druse remains.

Answer: Concerning point (1), we have initiated a new study to measure the size of crystals and druses in different parts of numerous recently collected plants. The results will serve as a reference catalog for comparison with fossil material. Identification of CaOx traces and the inclusion of statistical analyses will require a larger sample of fossil leaves, which is part of our planned future research.

Statistics on the present results appear to be of little use due to the frequent inhomogeneity of the samples.

(2) Answer: We have included the distribution patterns in Table 2.

3. Related to the question (3), a discussion about the variation of composition inside each clade is needed.

Answer: Data on the compositions are shown in the Results, in Table 2, and are mentioned in the Discussion. Further discussions about the variations would be speculations.

4. In the discussion part, I want to see some words to exclude other possible interpretations of the granular structures, e.g. cell nuclei or special mesophyll cells.

Answer: We have mentioned some examples of structures which could form visible structures in fossil leaves that may resemble CaOx casts, but show different distribution or shape, and we mentioned pollen in plant fossils, which also look similar, and the discussion about their differences to CaOx casts in a previous publication.

Some minor points are here:

ABSTRACT:

1. Line 46, “These granular structures resembled patterns”, specify what pattern please.

Answer: „in morphology and distribution“ added.

2. Line 47, “can be interpreted”, you can directly say “were interpreted”.

Answer: Ok, replaced.

3. Line 52, “the correlations still need to be investigated” can be deleted.

Answer: It has been deleted.

INTRODUCTION:

4. Line 62, here the proportion of angiosperms is listed, what about gymnosperms?

Almost all of the studies focused on higher plants and only few on gymnosperms. We have not found comprehensive statistics on the frequency of CaOx in gymnosperms, but we have included our experiences, that almost all Cycads and Conifers have CaOx crystals.

5. Line 109, “the obvious instability” necessarily needs references or detailed explanation!

Answer: Explanation added.

6. Lines 118 – 121, I cannot understand why these patterns are “confident” and “verifiable” identifiers, give references please.

Answer: „confident“ replaced by „reliably“; Reference added.

7. Lines 158 – 160, the sentence “In leaves of …” should not be put here but within the introduction part of living gymnosperm CaOx.

Answer: It has been deleted, as it was redundant.

DISCUSSION:

8. Line 571, “frequently” should be replaced by a certain proportion from your results.

Answer: Examples (Figs. 3-5) are mentioned; „frequently“ replaced.

9. Line 579, need references or images for support.

Answer:

10. Lines 580–581, this description of a previous study should be moved to the introduction part.

Answer: We have deleted this sentence.

11. There are still some spelling mistakes in the text, please check them.

Reviewer #2: Compared to the previous version of the manuscript, the addition of a table and several changes in the figures have improved the quality of the data, enhancing the relation between data and conclusions. The abstract and introduction are also more informative but to my points of views, there are still some points that need to be addressed.

Major issues:

General:

1. For me, the revised manuscript and the revised manuscript with track changes are different! Particularly, the results paragraphs on Eckfeld and Willerhausen collections differ. In my opinion, the ‘revised’ version seems more completed.

Answer: I have checked the files on my computer and could not find such differences. Perhaps a previous working version of the „manuscript with track changes“ had been submitted accidently. We would like to apologize for this.

2. The goal of the authors is much clearer and the work recentered on the recognition of CaOx druses. However, a hook to understand their importance is still missing (for instance in the abstract) and impedes to figure out the significance of your work. It’s unfortunate.

Answer: We have added a sentence about the importance at the end of the abstract.

3. The authors state that pyrite framboids are substitutes of former CaOx druses. This statement is interesting but remains hypothetical: the size of pyrite framboids can go up to several hundreds of micrometers, contrary to the size of granules observed here in the fossil leaves (60 µm max. according to table 2).

Several of the sentences would benefit from a more precautious approach (L321-324 of ‘revised version’ – not in track changes one + L524-525).

Answer: We added some notes in the discussion.

Material:

A response to a former question about the number of samples and specimens observed was directly added at the end of the material paragraph. It seems a bit clumsy and should be rephrased.

Answer: We have mentioned „the inhomogeneity of the samples“ as a difficulty to obtain statistically relevant data.

Results:

Consider moving the paragraph from l.280 to 291 to the method section (as it has been done for a former sentence), since it deals about the methodology of identification. Like this, and although it allows to understand your thought process, it is a bit hard to read, between two pieces of results.

Answer: Ok, we deleted this section, as it was redundant with parts in Introduction and Discussion.

The authors stress that in fossil leaves, ‘there was a high degree of uncertainty in the morphological interpretation’ (l281) while they latter state that ‘spherical morphology of suitable size’ (l294) is among the characteristic trends to identify Caox druses. Perhaps a consensus approach would be to state that in well-preserved fossils, morphology + distribution are the key characteristics but with poor preserved fossils, only the distribution was used?

Answer: Thanks for this suggestion. We have included it.

Discussion:

I recommend moving the second sentence later in the text (after ref. 19 and 20) so the occurrence of druses in old fossil plants stresses their importance in providing information on evolution, etc …

Answer: It has been done

Minor issues

- Typos: ‘whithout’, ‘staightforward’

- L42, compared with biomineral patterns in leaves -> compared with biomineral patterns in extant leaves?

Answer: It has been done

- L48, depends on varying fossilization conditions -> depends on fossilization conditions?

Answer: It has been done.

- L178, leaf fossils -> fossil leaves?

Answer: It has been done

- L458, ‘colour’ has been replaced by ‘composition’. But in the following, you’re definitively talking about colour since you mention ‘pale’ or ‘darker’ patches. Why not ‘by colour and composition (and texture)’?

Answer: Good suggestion; we agree.

---

## [Editor Report · Decision Letter 2]

31 Aug 2025

Detection of traces of calcium oxalate druses in fossil leaves of angiosperms and gymnosperms from different sites and geological periods

PONE-D-25-01413R2

Dear Dr. Malekhosseini,

We’re pleased to inform you that your manuscript has been judged scientifically suitable for publication and will be formally accepted for publication once it meets all outstanding technical requirements.

Kind regards,

Borja Cascales-Miñana

Academic Editor

PLOS ONE
---

## [Editor Report · Acceptance letter]

PONE-D-25-01413R2

PLOS ONE

Dear Dr. Malekhosseini,

I'm pleased to inform you that your manuscript has been deemed suitable for publication in PLOS ONE. Congratulations! Your manuscript is now being handed over to our production team.

Kind regards,

on behalf of

Dr. Borja Cascales-Miñana

Academic Editor

PLOS ONE